# Prefoldin complex promotes interferon-stimulated gene expression and is inhibited by rotavirus VP3

Yinxing Zhu [1], Yanhua Song[2,3], Dilip Kumar [4,6], Peter K. Jackson [2], B. V. Venkataram Prasad [4,5] & Siyuan Ding [1] ✉

Timely induction of interferons and interferon-stimulated genes (ISGs) is critical for successful host defense against viral infections. VP3 encoded by rotavirus is implicated in interferon antagonism. However, the precise mechanisms remain incompletely understood. By conducting tandem-affinity purification coupled with high-resolution mass spectrometry, we identify the prefoldin complex as the top cellular binding partner of VP3. Rotavirus infection is significantly enhanced in prefoldin subunit knockout cells. Using proteome-wide label-free quantification, we find that prefoldin assists in folding ubiquitin-like-modifier-activating-enzyme-3 (UBA3), both of which positively regulate ISG expression. Through direct and competitive binding, VP3 interferes with the chaperone activity of prefoldin, leading to unstable UBA3, reduces IRF9, and suppresses ISG transcription. Our findings report a novel function of a prefoldin-UBA3-IRF9-ISG axis in antiviral immunity and uncover new aspects of virus-host interactions that could be exploited for broad-spectrum antiviral therapeutic development.

In eukaryotes, a canonical prefoldin (PFDN) hexamer is composed of two different α subunits (encoded by *VBP1* also called *PFDN3*, and *PFDN5*) and four different β subunits (encoded by *PFDN6*, *PFDN4*, *PFDN2*, and *PFDN1*)[1]. Functionally, PFDN is a cytoplasmic chaperone complex that assists in folding actin and tubulin monomers and transferring them to chaperone proteins (TRiC/CCT) to form higher order structures[2–7]. However, PFDN5 is also found in the nucleus, where it binds to c-Myc (MYC) and transcription intermediary factor 1-beta, and suppresses MYC transcription activity[8,9]. Recent studies suggest that the PFDN and CCT complexes facilitate mammalian orthoreovirus replication by helping to fold its outer capsid protein σ3[10,11]. PFDN3, in complex with Cul2/VHL, is required for HIV-1 infection at a step between integrase-dependent proviral integration into the host genome and transcription of viral genes[12]. Hepatitis C virus (HCV)

F protein interacts with PFDN2, contributing to viral persistence in chronic HCV infection[13]. In all of these cases, PFDN seems to be pro-viral. Whether PFDN possesses antiviral function and regulates host innate immunity remains unexplored.

Rotaviruses (RVs) are the most common pathogens causing severe gastroenteritis and diarrhea in the young of many mammalian species, causing the death of more than 215,000 children per year[14,15]. Although NSP1 is an important virally encoded factor to inhibit interferon (IFN) signaling in RV-infected cells[16–18], VP3 also participates in innate immune evasion. In addition to its primary function as the viral mRNA capping enzyme[19–21], the N-terminal domain of VP3 mediates the degradation of mitochondrial antiviral signaling protein (MAVS), a key adaptor protein in the cytosolic RNA sensing pathway upstream of IFN induction[22,23]. VP3 also encodes a C-terminal 2′−5′-phosphodiesterase

[1]Department of Molecular Microbiology, Washington University School of Medicine, St. Louis, MO, USA. [2]Department of Microbiology and Immunology, Stanford School of Medicine, Stanford, CA, USA. [3]Institute of Veterinary Medicine, Jiangsu Academy of Agricultural Sciences, Nanjing, China. [4]Verna and Marrs McLean Department of Biochemistry and Molecular Pharmacology, Baylor College of Medicine, Houston, TX, USA. [5]Department of Molecular Virology and Microbiology, Baylor College of Medicine, Houston, TX, USA. [6]Present address: Trivedi School of Biosciences, Ashoka University, Sonepat, Haryana, India. ✉e-mail: siyuan.ding@wustl.edu

(PDE) domain to counteract OAS-RNase L-mediated antiviral signaling[24–26]. However, the precise gene-function relationship of VP3 in the viral replication cycle has not been fully elucidated, particularly in the context of the newly available RV reverse genetics system. In this study, we report on the host protein interactome of RV VP3 and elucidate a novel function of PFDN in promoting antiviral IFN signaling that VP3 counters during RV infection.

## Results

### VP3 binds to multiple subunits of the PFDN complex

To gain deeper insights into the biological functions of RV VP3 protein, we employed a quantitative proteomics approach to construct a comprehensive interactome network of host factors that co-precipitate with VP3. We adopted the G-LAP-Flp strategy, previously used in the interactome analysis for RV NSP1[27] and VP4[28], by fusing a localization and affinity purification (LAP) tag, i.e., a GFP-TEV-S-peptide, to the N-terminus of VP3 derived from the rhesus RV RRV strain. Subsequently, we generated a doxycycline-inducible HEK293 stable cell line expressing LAP-tagged VP3. Following tandem-affinity purification and TEV protease cleavage to remove the GFP tag, we separated eluted proteins using SDS-PAGE and visualized both VP3 itself and several bands corresponding to host proteins by silver staining (Supplementary Fig. 1). We then employed liquid chromatography mass spectrometry (LC-MS/MS) and rigorous bioinformatics analysis to identify high-confidence host factors interacting with VP3 (Supplementary Data 1). Five of the eight proteins identified were members of the PFDN complex (Fig. 1a). We conducted immunoprecipitation experiments to validate our mass spectrometry results. By co-transfection of each of the individual PFDN subunits and GFP-VP3 in HEK293 cells, we found that VP3 interacted with PFDN2-5, with the strongest interaction observed between VP3 and two PFDN α subunits, i.e., PFDN3 and PFDN5 (Fig. 1b). Using lysates from HEK293 inducible cell lines expressing GFP and GFP-tagged RV proteins VP2 and VP3, we found that VP3, but not GFP and VP2, specifically co-precipitated with endogenous PFDN3 (Fig. 1c). We further confirmed VP3-PFDN3 interaction by immunoprecipitation in RV-infected A549 cells that are interferon-competent (Fig. 1d). A His pull-down assay with purified recombinant VP3[21] and PFDN3 proteins confirmed direct binding between VP3 and PFDN3, while no significant interaction was observed with BSA or CDPK1, a control toxoplasma protein[29] (Fig. 1e). Bio-layer interferometry (BLI) assay further supported strong binding between PFDN3 and VP3 (Fig. 1f). In summary, these results indicate that VP3 directly interacts with the PFDN complex.

### PFDN knockout cells support increased RV replication

To investigate the functional role of PFDN in virus infection, we generated two single clonal PFDN knockout HEK293 cell lines using the CRISPR/Cas9 system. Complete deletion of PFDN3 or PFDN4 was confirmed by western blot (Fig. 2a, b) and Sanger sequencing (Supplementary Fig. 2a, b). As PFDN is known to be a chaperone for folding actin monomers[5], we visualized the formation of actin filaments using phalloidin staining as a functional assay for measuring PFDN activity. Knockout of PFDN3 or PFDN4 notably inhibited actin polymerization, similar to the effect of the actin related protein 2/3 (Arp2/3) complex inhibitor CK-666[30] (Supplementary Fig. 3). Since PFDN and the CCT complex act to fold reovirus capsid protein σ3[10], we sought to use reovirus infection as an additional control. Consistent with the previous study[10], PFDN knockout significantly inhibited reovirus replication (Supplementary Fig. 4). Based on these results, we hypothesized that PFDN might also play an important role in folding RV VP3 and thus be essential for RV replication. However, PFDN3 or PFDN4 knockout did not reduce VP3 protein levels by microscopy and flow cytometry analysis (Supplementary Fig. 5a–c), suggesting that VP3 is unlikely to be a direct substrate for PFDN. Surprisingly, knockout of either PFDN3 or PFDN4 led to an increase in intracellular viral RNA levels of bovine

RV UK strain at 48 and 72 h post-infection (hpi) (Fig. 2c). PFDN3 or PFDN4 knockout also resulted in enhanced infectious RV titers at late time points, i.e., 48 and 72 hpi, but not at early time points, i.e., 1, 8, and 24 hpi (Fig. 2d). Moreover, we tested two other RVs, i.e., human RV WI61 strain and porcine RV OSU strain. The replication of both viruses was also higher in the absence of PFDN3 or PFDN4 (Fig. 2e, f). These results strongly suggest that PFDNs have a previously unknown antiviral role that inhibits the replication of multiple RV strains.

### PFDN promotes IFN-β-induced ISG expression

To investigate the mechanisms by which PFDN inhibits RV infection, we extended our analysis to vesicular stomatitis virus (VSV), a negative-sense single-stranded RNA virus unrelated to RV. After infecting cells with a recombinant VSV expressing GFP, we observed that PFDN knockout cells had significantly higher GFP levels than wild-type (WT) cells (Supplementary Fig. 6). This finding suggests that the PFDN complex inhibits viral infections beyond RVs. Since VSV is highly sensitive to IFNs[31], and the antiviral effect on RV was more pronounced at later time points (Fig. 2c, d), we reasoned that PFDN may regulate IFN and/or ISG levels to exert its antiviral activities. To explore this further, we stimulated both WT and PFDN knockout HEK293 cells with IFN-β and profiled the transcriptome using bulk RNA sequencing. In a previous study, PFDN5 was identified as a transcription repressor of MYC[8]. We confirmed that MYC expression increased in PFDN4 knockout cells and restored to homeostatic levels in PFDN4 knockout cells complemented with Flag-tagged PFDN4 (Fig. 3a and Supplementary Data 2). Notably, a large number of canonical ISGs showed an opposite trend and were substantially downregulated in PFDN4 knockout cells following IFN-β stimulation (Fig. 3a). Adding back Flag-tagged PFDN4 in knockout cells reversed this ISG reduction (Fig. 3a). In contrast, the mRNA levels of GAPDH and interleukin-11, a chemokine-induced by the NF-κB pathway[32], were not altered (Fig. 3a). Further RT-qPCR analysis validated that PFDN3 or PFDN4 deficiency significantly inhibited the transcription of multiple ISGs, including IFITM1, MX1, and OAS3 (Fig. 3b). This was independently validated in multiple single clonal PFDN3 and PFDN4 knockout cells (Supplementary Fig. 7a–c) and with additional ISGs such as OAS1 and ISG15 (Supplementary Fig. 8a). The inhibition was also reflected at the protein level for multiple ISGs (Supplementary Fig. 8b). Since MX1 and OAS3 are the most highly induced genes upon IFN stimulation, we used these two genes as representative ISGs for the following analysis. Rescuing PFDN3 and PFDN4 in respective knockout cells restored IFN-β-induced OAS3 mRNA level (Fig. 3b) and protein level (Fig. 3c). We also observed that PFDN4 was nearly undetectable in PFDN3 knockout cells and PFDN3 level was low in PFDN4 knockout cells, suggesting that PFDN complex was not stable after knocking out either PFDN3 or PFDN4 subunit (Fig. 3c). Indeed, PFDN3 and PFDN4 double knockout cells stimulated with IFN-β showed no further reduction in ISG expression compared to either single knockout cells, suggesting that the observed phenotypes are driven by lack of the entire PFDN complex (Supplementary Fig. 9). OAS3, in conjunction with RNase L, mediates global viral and host RNA degradation in response to cytosolic double-stranded RNA (dsRNA) sensing[33,34]. We next used poly (I:C), a viral dsRNA mimicry, as a functional assay to probe the effect of reduced OAS3 levels on the activation of the OAS-RNase L pathway. By monitoring ribosomal RNA degradation as a surrogate of RNase L activation, we found that PFDN4 knockout notably inhibited rRNA degradation induced by poly(I:C) transfection (Supplementary Fig. 10). In contrast, rRNA degradation induced by transfection of 2′–5′A, a second messenger that bypasses the OAS enzymes and directly activates RNase L[24,26,33–35], was not altered in PFDN4 knockout cells (Supplementary Fig. 10). Further, RV infection significantly induced ISG production in WT cells, but not in PFDN3 and PFDN4 knockout cells (Fig. 3d). Collectively, these results strongly suggest that PFDN plays a vital role in ISG production induced by type I IFN or viral infection.

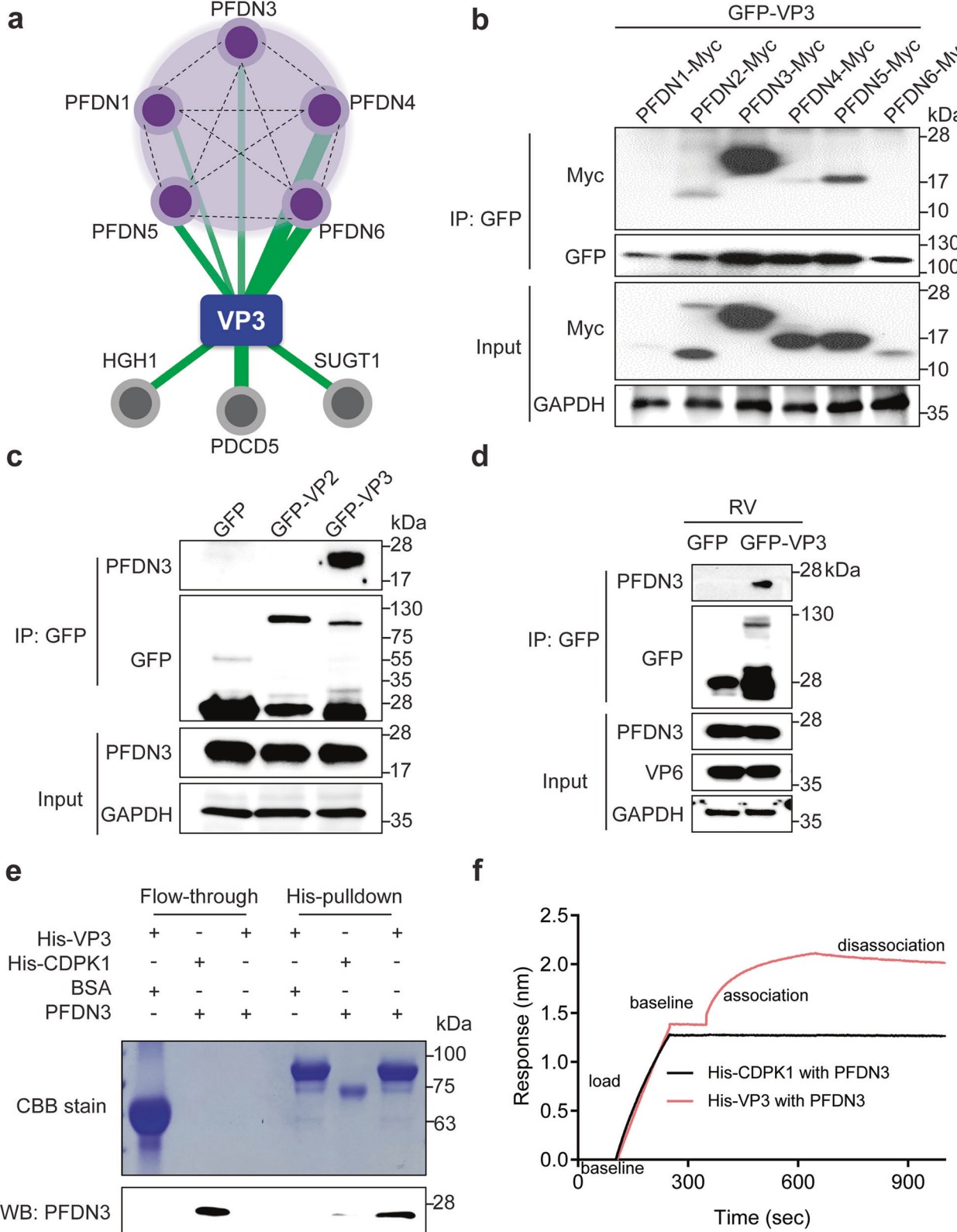

## UBA3 is a novel substrate of PFDN

Next, we sought to examine the molecular mechanisms by using an unbiased screening approach to identify new PFDN substrates likely responsible for regulating ISG levels. We profiled the global cellular proteome of WT, *PFDN3* knockout, and *PFDN4* knockout cells using label-free quantification (LFQ) mass spectrometry. As expected, PFDN3 and PFDN4 levels were among the top proteins with low

abundance in the knockout cells compared to WT cells (Fig. 4a, b, Supplementary Data 3), consistent with our western blot results (Fig. 3c). Notably, three additional proteins, ubiquitin-like modifier activating enzyme 3 (UBA3), isocitrate dehydrogenase 2 (IDH2), and aldehyde dehydrogenase 2 (ALDH2)−also exhibited substantial reduction in both *PFDN3* and *PFDN4* knockout cells (Fig. 4a, b). To elucidate the specific protein responsible for ISG production, we

**Fig. 1 | RV VP3 interacts with the PFDN complex. a** A protein-protein interactome of the bait protein VP3 (blue node) and high-confidence host interaction partners (purple and gray nodes). Solid green lines represent novel interactions identified in this study. The widths of the green lines correspond to the number of spectral peptides identified in the IP-MS experiment. Dotted black lines indicate curated protein-protein interactions in public proteomics databases. **b** Lysates of HEK293 cells co-transfected with GFP-VP3 and PFDN1-6-Myc were subject to immunoprecipitation using α-GFP antibody and probed with α-Myc antibody. **c** Lysates of doxycycline-induced HEK293 cells that expressed GFP, GFP-VP2, and GFP-VP3 were subject to immunoprecipitation using α-GFP antibody and probed with α-PFDN3 antibody. **d** A549 cells were transfected with the indicated plasmids for 24 h,

followed by infection with RV (RRV strain) at an MOI of 1 for 12 h. Co-immunoprecipitation was performed, and the interaction between VP3 and PFDN3 was analyzed by western blot. **e** A His-pulldown assay of VP3 and human PFDN3. His-VP3 and His-CDPK1 were incubated with $Ni^{2+}$ beads and then with BSA or PFDN3. Flow-through and His-tagged proteins in the eluates were detected by Coomassie blue (CBB) staining and examined by western blotting using a rabbit anti-PFDN3 antibody. **f** BLI sensorgrams obtained using biosensor loaded with His-tagged CDPK1 or VP3 (10 nM) (load) and incubated with 50 μM PFDN3 protein (association). The dissociation is shown on the right. The binding of PFDN3 to VP3 or CDPK1 is shown in red and black, respectively. For all figures, experiments were repeated at least three times. Experiments (**b**–**e**) were repeated at least two times.

treated cells with small-molecule inhibitors MLN4924[36], AGI6780[37], and Disulfiram[38], which block UBA3, IDH2, and ALDH2, respectively. Among the compounds tested, only MLN4924, an inhibitor of UBA3 and the neddylation pathway[39], recapitulated the reduced ISG expression observed in *PFDN* knockout cells (Supplementary Fig. 11). Notably, MLN4924 exerted strong inhibition of MX1 expression even at the lowest concentration tested (100 nM) (Fig. 4c). Furthermore, UBA3 protein levels were reduced in both *PFDN3* knockout and *PFDN4* knockout cells compared to WT HEK293 cells (Fig. 4d), suggesting that UBA3 is possibly a substrate of PFDN. Due to the lack of appropriate antibodies suitable for PFDN immunoprecipitation, we reconstituted *PFDN4* knockout cells with Myc-tagged PFDN subunits and found that endogenous UBA3 indeed co-precipitated with the PFDN complex but not the Myc tag empty vector (Fig. 4e). To investigate whether UBA3 directly interacts with PFDN, we measured the interaction between purified recombinant GST-UBA3 and His-PFDN3 protein, with His-CDPK1 protein as a control. The His-pull-down assay confirmed the direct interaction between UBA3 and PFDN3 (Fig. 4f). Additionally, from BLI, we demonstrated that UBA3 directly interacts with PFDN3 but not CDPK1 (Fig. 4g). These results indicate that UBA3 is a bona fide substrate of PFDNs.

## UBA3 is essential for ISG production

Because UBA3 serves as the substrate of PFDN, which positively regulates ISG expression, we investigated the role of UBA3 in ISG signaling. To explore this, we generated *UBA3* knockout HEK293 cells using two independent single-guide RNAs (sgRNAs) and confirmed the knockout efficiency by western blot (Fig. 5a). Additionally, we created *UBA3* and *PFDN4* double knockout cells. *PFDN4* knockout cells were chosen because of the more extensive characterization in our system, including available RNA-seq data (Fig. 3a) and rRNA degradation assay (Supplementary Fig. 10). Consistent with our hypothesis, *UBA3* knockout cells phenocopied the inhibited IFN-β-stimulated MX1 induction observed in *PFDN4* knockout cells, with no further decrease in MX1 inhibition in the double knockout cells (Fig. 5b), suggesting that these two proteins likely function within the same pathway. Consistent with previous data (Fig. 4d), UBA3 protein level was almost undetectable in *PFDN4* knockout cells (Fig. 5c). Importantly, OAS3 levels, like MX1, were not further reduced in *UBA3* and *PFDN4* double knockout cells compared to single knockout cells (Fig. 5c). Furthermore, we validated the pivotal role of UBA3 in ISG production induced by RV infection (Fig. 5d). To determine whether UBA3 restricts viral replication via the IFN pathway, we treated cells with a pan-JAK kinase inhibitor ruxolitinib to block ISG expression downstream of IFN signaling. While ruxolitinib treatment substantially enhanced RV replication in WT cells, this effect was lost in *UBA3* knockout cells (Fig. 5e). To identify the missing link between UBA3 and ISG induction, we examined STAT1, STAT2, IRF3, and IRF9, known as the interferon-stimulated gene factor 3 (ISGF3) complex[40]. While *PFDN3* or *PFDN4* knockout did not affect bulk protein levels of total STAT1, IRF3, phosphorylated STAT1, and STAT2, the levels of IRF9, a key component of the ISGF3 complex, were substantially lower (Supplementary Fig. 12). Knocking out UBA3 reduced mRNA expression of IRF9 in IFN-β stimulated

HEK293 cells (Fig. 5f). IRF9 protein level was also downregulated in UBA3 knockout cells or WT cells receiving MLN4924 treatment (Fig. 5g). These findings support the notion that PFDNs promote ISG expression through UBA3 and IRF9 and inhibit viral infections by stabilizing the novel cell substrate UBA3.

## VP3 competitively inhibits the interaction between PFDN and UBA3

Because PFDN directly interacts with both VP3 and UBA3 (Figs. 1f, 4g), we hypothesized that RV VP3 might compete with UBA3 for PFDN binding. In RV-infected cells, the interaction between UBA3 and PFDN decreased (Fig. 6a). Next, we aimed to identify the specific VP3 domain essential for its interaction with PFDN3. We took advantage of a series of VP3 mutants encoding different lengths of mitochondrial antiviral-signaling protein (MAVS) binding, N7-MTase, 2′-O-MTase, GTase/RTPase, and PDE domains[22] (Fig. 6b). Immunoprecipitation assays mapped the PFDN binding site within VP3 to a small region (137 amino acids) that corresponds to the GTase/RTPase function (Fig. 6c). Upon a closer examination of the predicted structure of PFDN3 and VP3 complex using AlphaFold-3, we identified a 20-amino acid region (670-689) as the potential interaction interface (Fig. 6d). To experimentally test this prediction, we generated a VP3 mutant lacking this region (Δ670-689). Compared to the full-length VP3, the deletion of amino acids 670-689 drastically reduced the interaction with PFDN3 (Fig. 6e). Further, a chemically synthesized 670-689 VP3 peptide effectively abolished the interaction between VP3 and PFDN3 in the BLI assay (Fig. 6f). Furthermore, we aligned VP3 sequences of various RV strains from different species[41] and identified two highly conserved residues, 671D and 677Y (Fig. 6g). To directly assess the importance of these two amino acids, we introduced point mutations in VP3 and generated a D671A and Y677A double mutant. Immunoprecipitation confirmed that this mutant lost its binding affinity with PFDN3 (Fig. 6h). To further validate the interaction between the VP3-GTase domain and PFDN3 during viral infection, we rescued a recombinant RV expressing Flag-tagged GTase/RTPase domain of VP3. The recombinant virus replicated comparably to the parental RV in MA104 cells (Supplementary Fig. 13a), and the genetically edited gene segment 7 that encodes NSP3, a 2 A peptide from porcine teschovirus-1 (P2A), and Flag-tagged GTase/RTPase domain of VP3 was confirmed by Sanger sequencing (Supplementary Fig. 13b). Due to the lack of VP3 antibodies, we rescued a Flag-tagged VP3-GTase domain virus. Co-immunoprecipitation assays confirmed that the Flag-tagged VP3-GTase domain interacted with PFDN3 under infection conditions (Fig. 6i). Overall, our results highlight a novel role of VP3 in the RV-host arms race through competitive binding with the PFDN complex.

## D671A/Y677A is critical for ISG suppression and RV infection

Using our recently optimized reverse genetics system[42], we rescued recombinant RVs encoding either WT or mutant VP3 proteins and performed infection experiments in both WT and *PFDN* knockout cells. The mutations of D671A and Y677A were confirmed in the rescued virus stocks by Sanger sequencing (Supplementary Fig. 13c). Consistent with the ectopic expression data, infection of RV with D671A

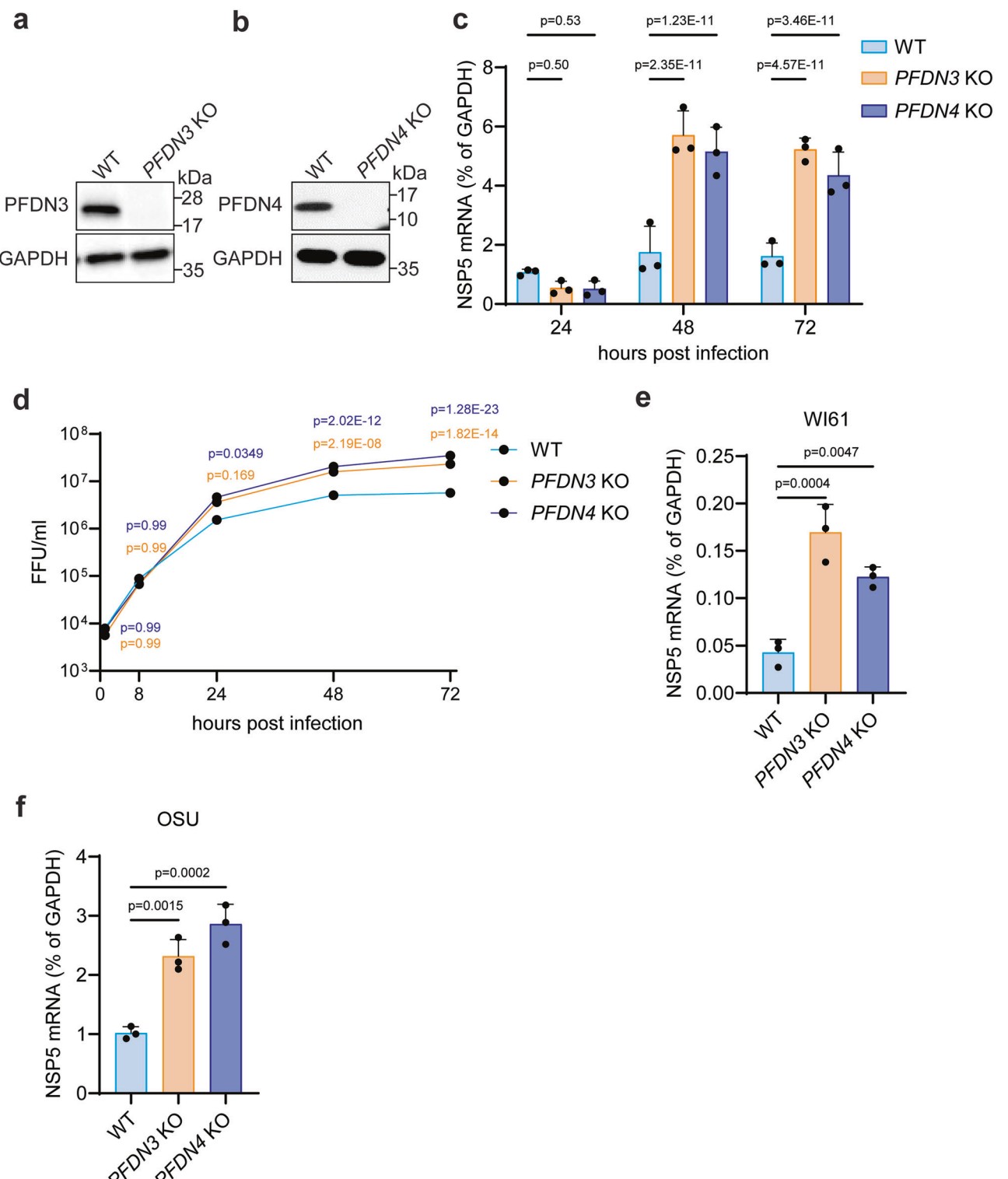

**Fig. 2 | CRISPR knockout of *PFDN* promotes RV replication. a** Western blot detection of PFDN3 in knockout (*PFDN3* KO) and wild-type (WT) HEK293 cells. **b** Western blot detection of PFDN4 in knockout (*PFDN4* KO) and WT HEK293 cells. **c** WT, *PFDN3* KO, and *PFDN4* KO HEK293 cells were infected with RV (UK strain, MOI = 5), and viral NSP5 mRNA level was measured at 24, 48, and 72 hpi by RT-qPCR. Data represents the average of three experiments; error bars indicate SEM (two-way ANOVA with Tukey's multiple comparisons test). **d** WT, *PFDN3* KO, and *PFDN4* KO HEK293 cells were infected with RV (UK strain, MOI = 5), and viral titers were measured at 1, 8, 24, 48, and 72 hpi by an FFU assay. Data represents the average of three experiments; error bars indicate SEM (two-way ANOVA with Dunnett's multiple comparisons test). **e** WT, *PFDN3* KO, and *PFDN4* KO HEK293 cells were infected with RV (WI61 strain, MOI = 5), and viral NSP5 mRNA level was measured at 72 hpi by an FFU assay. Data represents the average of three experiments; error bars indicate SEM (one-way ANOVA with Dunnett's multiple comparisons test). **f** WT, *PFDN3* KO, and *PFDN4* KO HEK293 cells were infected with RV (OSU strain, MOI = 5), and viral NSP5 mRNA level was measured at 72 hpi by an FFU assay. Data represents the average of three experiments; error bars indicate SEM (one-way ANOVA with Dunnett's multiple comparisons test).

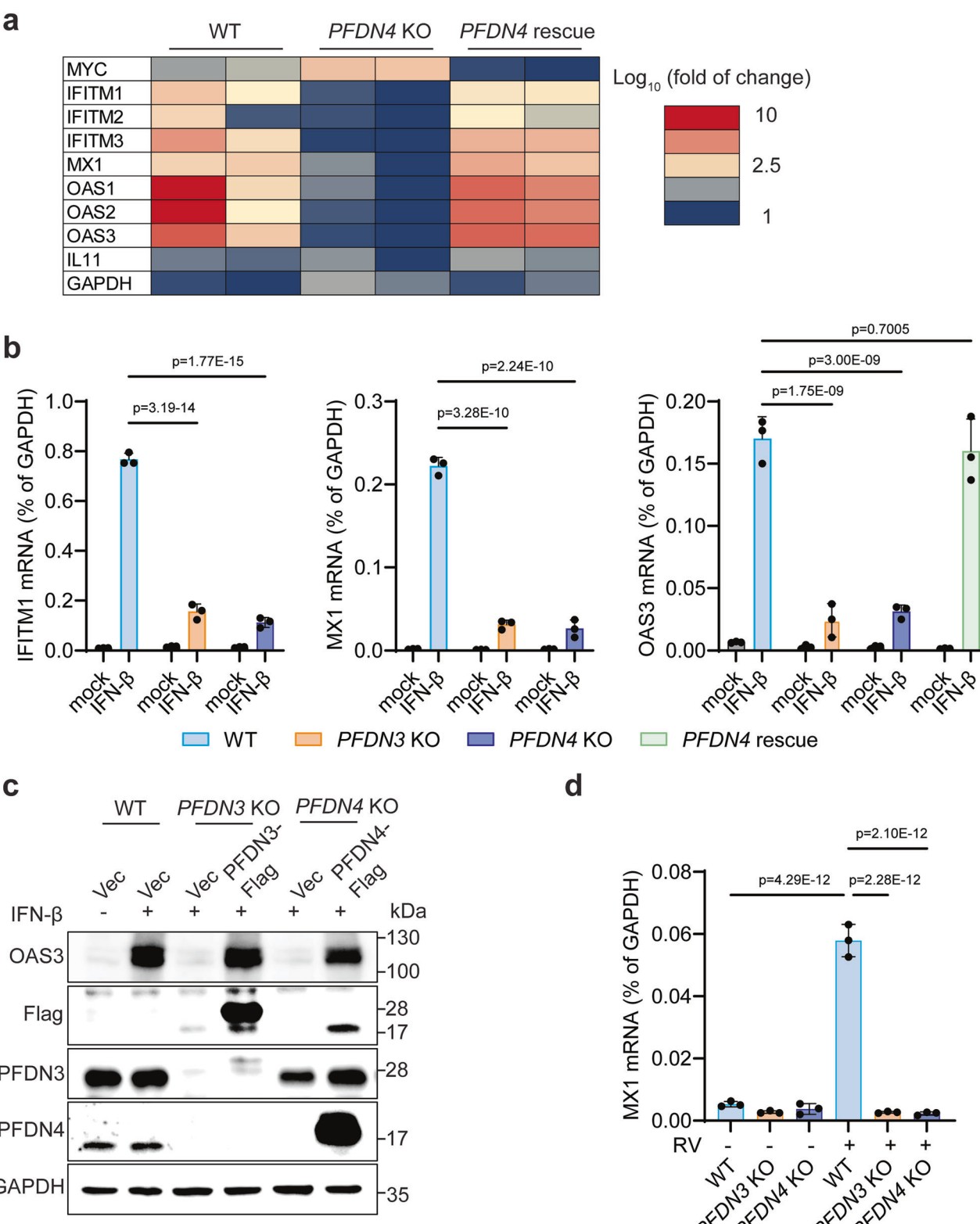

and Y677A mutations in VP3 resulted in higher levels of MX1 expression in WT cells compared to infection of the parental viral strain (Supplementary Fig. 14a). This defect was abolished in either *PFDN3* or *PFDN4* knockout cells where the ISG levels were readily inhibited (Supplementary Fig. 14a).

We next compared parental and mutant virus replication in donor-derived human intestinal enteroids, which consist of primary human intestinal epithelial cells[43]. Differentiated enteroid monolayers in 2D polarized transwells were infected with either parental RV or the VP3-D671A/Y677A mutant. While no significant difference was observed at 8 hpi, the mutant virus showed attenuation at 48 hpi (Supplementary Fig. 14b). In parallel, infection with the VP3-D671A/Y677A mutant induced higher MX1 levels than parental virus (Supplementary Fig. 14c), supporting a role for D671 and Y677 in modulating ISG responses. Nevertheless, because viral replication at early time points is minimal, these data do not allow us to definitively

**Fig. 3 | PFDN mediates IFN-β- and RV-induced ISG production. a** HEK293 cells (WT, *PFDN4* KO, and *PFDN4* rescue) were stimulated with IFN-β (500 U/ml) for 24 h. The *PFDN4* rescue cells were *PFDN4* KO cells transduced with Flag-tagged PFDN4 (single clone). A heatmap summarizes the transcript levels of MYC, ISGs (IFITM1, IFITM2, IFITM3, MX1, OAS1, OAS2, and OAS3), IL11, and GAPDH from RNA-sequencing data. The colors represent the $Log_{10}$ (fold change) in gene counts identified in the RNA-sequencing experiments. **b** WT, *PFDN3* KO, and *PFDN4* KO cells were stimulated with IFN-β (500 U/ml), and ISGs (IFITM1 and MX1) mRNA level was measured at 24 hpi by RT-qPCR (left and middle panel). WT, *PFDN3* KO, *PFDN4* KO, and *PFDN4* rescue cells were stimulated with IFN-β (500 U/ml), and the OAS3 mRNA level was measured at 24 hpi by RT-qPCR (right panel). Data represents the average of three experiments; error bars indicate SEM (two-way ANOVA with Dunnett's multiple comparisons test). **c** WT, *PFDN3* KO, *PFDN4* KO HEK293 cells were transduced with vector or PFDN3-Flag or PFDN4-Flag plasmid for 24 h and stimulated with IFN-β (500 U/ml) for another 24 h. Total cell lysates were harvested and examined by western blot with indicated antibodies. **d** WT, *PFDN3* KO, and *PFDN4* KO HEK293 cells were infected with or without RV (RRV strain) at an MOI of 5 for 24 h. MX1 mRNA level was measured at 24 hpi by RT-qPCR, which was normalized to GAPDH. Data represents the average of three experiments; error bars indicate SEM (two-way ANOVA with Šídák's multiple comparisons test). For all figures, experiments were repeated at least three times.

determine which stage of infection is most affected by the VP3 mutations.

We also found that the ability of VP3 to hijack PFDN and dampen ISG induction could be pharmacologically mimicked. Treatment of human intestinal enteroids with cytochalasin D (CytoD), an inhibitor of actin polymerization[44], induced high ISG expression, which was partially reversed by the neddylation inhibitor MLN4924 (Supplementary Fig. 14d). Finally, VP3 mutations contributed to reduced replication in a neonatal mouse model compared with the parental RV (Supplementary Fig. 14e, f). Together, these results suggest that the "sponge effect" of RV VP3 on PFDN may be generalized to conditions that cause actin monomer accumulation, thereby limiting cytoplasmic PFDN availability.

## Discussion
PFDN has been extensively characterized for its role in the folding of actin and tubulin. However, our study reveals a previously unrecognized function of PFDN in host defense against viral invasion. We have demonstrated that knocking out PFDN3 or PFDN4 subunits significantly enhances viral replication (Fig. 2), indicating its crucial role in antiviral defense. We also show that PFDN operates upstream of UBA3 (Fig. 4). Our working model is summarized in Fig. 7. Based on our data, we propose that PFDN stabilizes UBA3, thereby facilitating the expression of ISGs through IRF9, which is essential for inhibiting viral replication. The absence of PFDN disrupts this stabilization, leading to a decrease in ISG production and enhanced RV infection (Fig. 5). From the rotavirus infection perspective, we identified by immunoprecipitation that the viral protein VP3 competes with PFDN for binding to UBA3 (Fig. 6). This competitive binding by VP3 disrupts the PFDN-UBA3 interaction, resulting in the suppression of ISG production and highlighting a novel strategy employed by RVs to evade the host immune response. We mapped the PFDN binding site within VP3 to a specific 137-amino acid region within the GTase/RTPase domain to further elucidate the interaction dynamics. Structural predictions using AlphaFold-3 identified a 20-amino acid region (670-689) as the potential interaction interface. Experimental validation through the generation of a VP3 mutant lacking this region (Δ670-689) demonstrated a drastic reduction in interaction with PFDN. Additionally, point mutations at conserved residues 671D and 677Y within this region confirmed their critical role in the VP3-PFDN interaction (Fig. 6 and Supplementary Fig. 14). While our data suggest these residues are important for ISG inhibition, additional experiments will be required to determine whether the replication phenotype of the VP3 mutants is exclusively due to disruption of PFDN antagonism or to other VP3 functions.

Our IP-MS data reveal that VP3 interacts with the PFDN complex (Fig. 1a). However, we did not detect previously reported interaction partners such as MAVS and OAS3[22-26]. The absence of MAVS and OAS3 in the HEK293 overexpression pulldown likely reflects limitations of this system, as certain interactions may only occur under infection conditions or in a cell type dependent manner.

The precise mechanism by which UBA3 controls IRF9 production remains unknown. Our results indicate that PFDN knockout does not affect STAT1 and STAT2. However, PFDN knockout leads to a decrease in IRF9 protein levels (Supplementary Fig. 10). This observation suggests that PFDN may play a role in stabilizing or regulating IRF9, which is crucial for ISG production. Previous studies have highlighted the importance of the neddylation pathway in immune responses, particularly noting that neddylation is critical for the modification and activation of IRF7[45]. Our study finds that UBA3 regulates IRF9 possibly through neddylation modification and stability. This hypothesis is supported by the observed reduction in IRF9 levels upon PFDN knockout and IRF9 levels reduced by MLN4924 treatment (Fig. 5f)[46]. To further elucidate this potential connection, additional studies are required to investigate whether UBA3 directly controls IRF9 neddylation. Such studies could involve examining the neddylation status of IRF9 in the presence and absence of UBA3, as well as assessing the impact of UBA3 on the stability and function of IRF9. Understanding this relationship could provide deeper insights into the regulatory mechanisms governing ISG production and the broader antiviral immune response.

Although previous studies have suggested a pro-viral role for PFDN[10,13], our results demonstrate that PFDN exhibits anti-RV activity. This discrepancy may be attributed to virus-specific functions of PFDN; in other viruses, PFDN may play a more critical role in assisting viral protein folding, whereas in our system, its impact on ISG expression appears to be predominant.

In summary, our findings broaden the current understanding of virus-host interactions, report a novel role of PFDN in antiviral innate immunity, and offer potential targets for therapeutic intervention. Whether this competitive inhibition by VP3 to block UBA3 represents a conserved microbial strategy to undermine the host defense mechanism remains to be explored.

## Methods
### Cell culture and viruses
HEK293 and A549 cells were cultured in DMEM (Sigma-Aldrich) supplemented with 10% heat-inactivated fetal bovine serum (FBS). HT-29 cells were cultured in Advanced DMEM (Sigma-Aldrich) supplemented with 10% heat-inactivated fetal bovine serum (FBS). MA104 cells (CRL-2378, ATCC) were cultured in Medium 199 (M199, Sigma-Aldrich) supplemented with 10% heat-inactivated fetal bovine serum (FBS), 100 I.U. penicillin/ml, 100 μg/ml streptomycin, and 0.292 mg/ml L-glutamine. The BHK-T7 cell line was kindly provided by Dr. Ursula Buchholz (Laboratory of Infectious Diseases, NIAID, NIH, USA)[47] and cultured in complete DMEM supplemented with 0.2 μg/ml G-418 (Promega) every other passage. *SERPINB1* knockout MA104 cells for virus rescuing were cultured in complete M199[42]. Human small intestinal enteroids (H549) were cultured in an organoid culture medium.

The RV strains used in this study include UK[48], OSU[49], WI61[50], rSA11-RRV-VP3, rSA11-RRV-VP3-671A/677 A, and rSA11-GTase-Flag propagated in MA104 cells. Before infection, all RV inoculates were activated with 5 μg/ml of trypsin (Gibco Life Technologies, Carlsbad, CA) for 30 min at 37 °C. VSV-GFP virus was propagated in Vero cells.

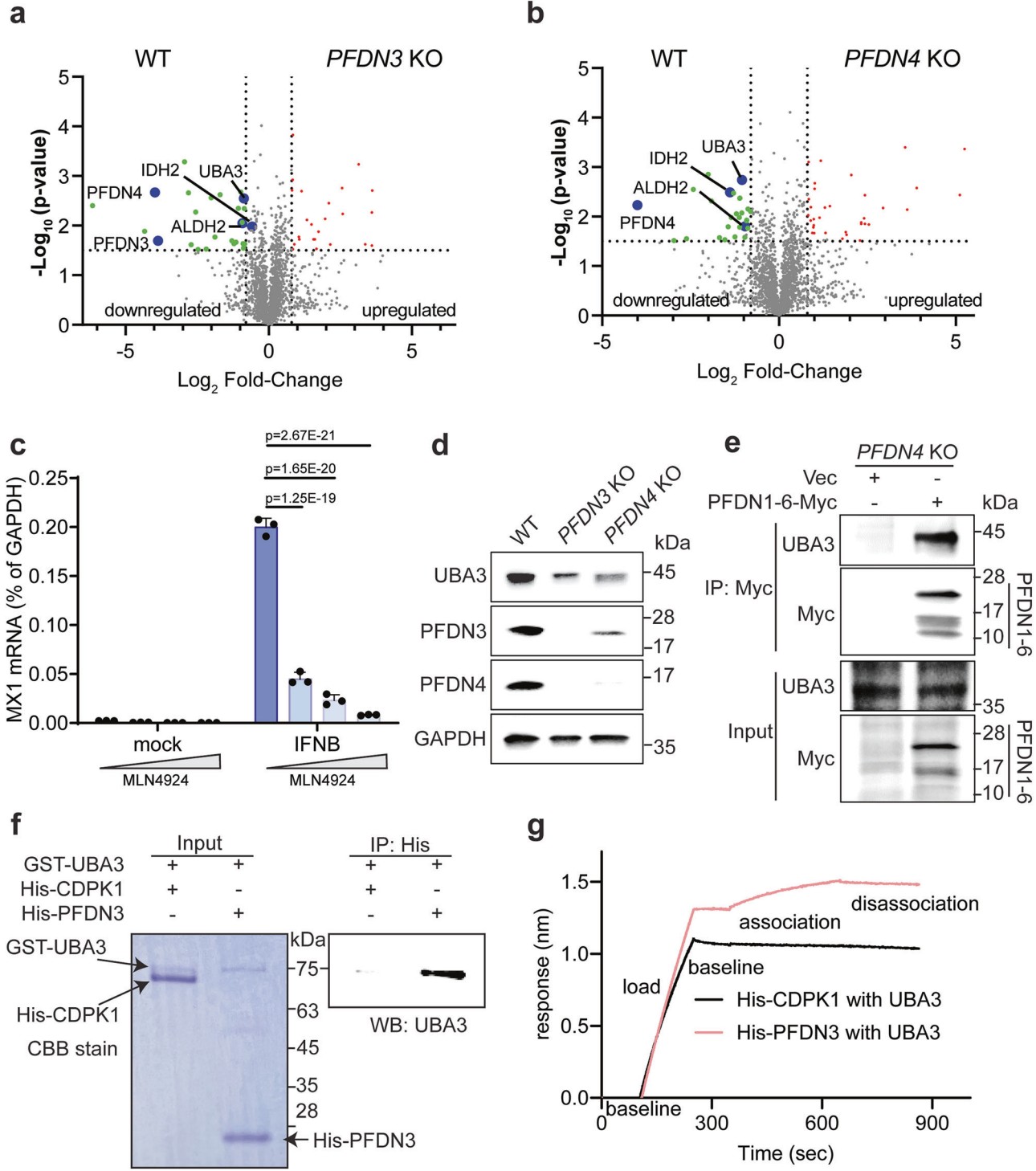

**Fig. 4 | UBA3 is the substrate of PFDN and activates ISG production. a** Volcano plot of mass spectrometry data from WT and *PFDN3* KO cells. Proteins downregulated in both *PFDN3* KO and *PFDN4* KO cells were shown in blue dots. Statistically significant proteins were identified using a two-sided *t* test with a permutation-based FDR set at 0.05. **b** Volcano plot of mass spectrometry data from WT, and *PFDN4* KO cells. Proteins downregulated in both *PFDN3* KO and *PFDN4* KO cells were shown in blue dots. Statistically significant proteins were identified using a two-sided *t* test with a permutation-based FDR set at 0.05. **c** HEK293 cells were treated with increasing concentrations of MLN4924 (0, 0.1 μM, 1 μM, and 10 μM) for 24 h and stimulated with or without IFN-β (500 U/ml) for another 24 h. MX1 mRNA was measured by RT-qPCR. Data represents the average of three experiments; error bars indicate SEM (one-way ANOVA with Dunnett's multiple

comparisons test). **d** WT, *PFDN3* KO, and *PFDN4* KO HEK293 cell lysates were harvested and examined by western blot with the indicated antibodies. **e** *PFDN4* KO HEK293 cells were transfected with PFDN1-6-Myc plasmids for 24 h. Cell lysates were subject to immunoprecipitation using α-Myc antibody and probed for α-UBA3 antibody. PFDN3 is the top band, and PFDN1-4 and PFDN6 are similar in size, as shown on the bottom. **f** A His-pulldown assay of PFDN3 and UBA3. His-CDPK1 or His-PFDN3 were incubated with Ni²⁺ beads and then incubated with GST-UBA3. UBA3 protein in the eluates was detected by western blotting using a mouse anti-UBA3 antibody. **g** BLI sensorgrams were obtained using a biosensor loaded with His-tagged CDPK1 or PFDN3 (50 nM) (load) and incubated with 50 μM UBA3 protein (association). The dissociation is on the right. The binding of PFDN3 to UBA3 is shown in red. For **d**–**f**, experiments were repeated at least three times.

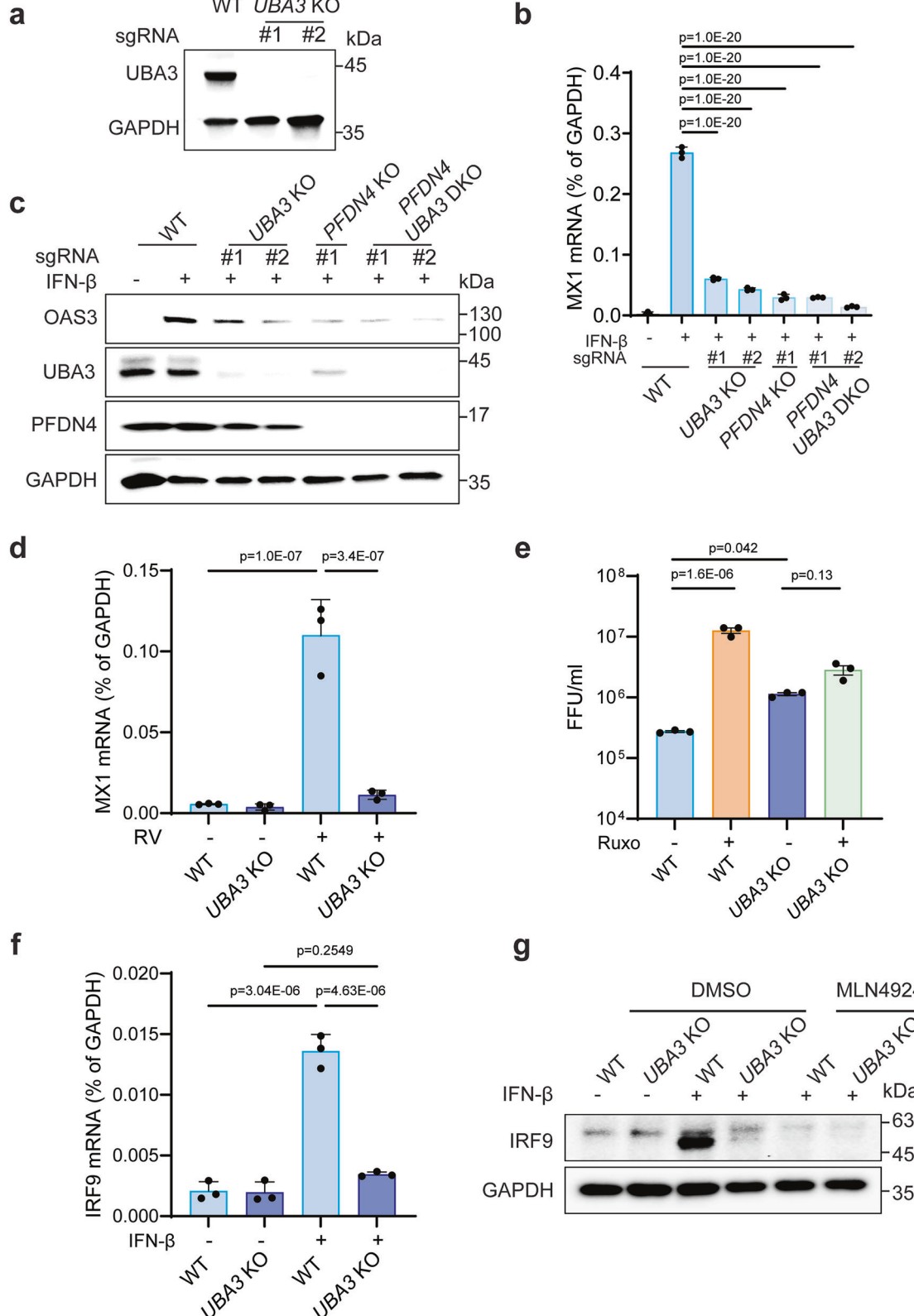

## Plasmid construction

EGFP-S-tandem-tagged RRV VP3 and VP2 in the LAP6 vector have been previously described[27]. Myc-Flag-tagged PFDN1-6 constructs were purchased from Origene (RC200584, RC204553, RC208482, RC203137, RC209347, and RC207901, respectively). VP3 truncations were cloned into EGFP-C1 vector[22]. GFP-VP3 Δ670-689 and GFP-VP3-D671A/Y677A plasmids were generated using a Quikchange II site-

directed mutagenesis kit (Agilent). The RV rescue plasmids: pT7-SA11-VP1, pT7-SA11-VP2, pT7-RRV-VP3, pT7-SA11-NSP4, pT7-SA11-VP4, pT7-SA11-VP6, pT7-SA11-VP7, pT7-SA11-NSP1, pT7-SA11-NSP2, pT7-SA11-NSP3, and pT7-SA11-NSP5 were prepared as described previously[51]. The C3P3-G3 plasmid was kindly provided by Dr. Philippe Jais (Eukarÿs)[42,52,53]. To generate pT7-RRV-VP3-D671A/Y677A plasmid, GAT (2011-2013) was mutated to GCT (2011-2013), and TAT (2029-2031)

**Fig. 5 | UBA3 activates ISG production. a** Western blot detection of UBA3 and GAPDH in WT and *UBA3* KO (sgRNA1, sgRNA2) HEK293 cells. Experiments were repeated two times. **b** WT, *UBA3* KO (sgRNA1, sgRNA2), *PFDN4* KO, *PFDN4* and *UBA3* double KO (sgRNA1, sgRNA2) HEK293 cells were stimulated with IFN-β (500 U/ml) for 24 h. MX1 mRNA level was measured by RT-qPCR. Data represents the average of three experiments; error bars indicate SEM (one-way ANOVA with Dunnett's multiple comparisons test). **c** WT, *UBA3* KO (sgRNA1, sgRNA2), *PFDN4* KO, and *PFDN4* and *UBA3* double KO (sgRNA1, sgRNA2) HEK293 cells were stimulated with IFN-β (500 U/ml) for 24 h. Cell lysates were analyzed by western blot. **d** WT, *UBA3* KO HEK293 cells were infected with or without RV (RRV strain, MOI of 5) for 24 h. MX1 mRNA level was measured by RT-qPCR. Data represents the average of three experiments; error bars indicate SEM (two-way ANOVA with Tukey's multiple comparisons test). **e** WT and *UBA3* KO HEK293 cells were treated with or without Ruxo (10 μM) and IFN-β (500U/ml) for 24 h, and then infected with UK (MOI = 5) for 72 h. Virus titers were determined by an FFU assay. Data represents the average of three experiments; error bars indicate SEM (one-way ANOVA with Tukey's multiple comparisons test). **f** WT and *UBA3* KO HEK293 cells were induced with IFN-β (500 U/ml) for 24 h. IRF9 mRNA level was measured by RT-qPCR. Data represents the average of three experiments; error bars indicate SEM (one-way ANOVA with Tukey's multiple comparisons test). **g** WT and *UBA3* KO HEK293 cells were treated with DMSO or MLN4924 (1 μM) for 24 h, then induced with IFN-β (500 U/ml) for 24 h. Cell lysates were analyzed by western blot.

was mutated to GCT (2029-2031) using Quikchange II site-directed mutagenesis kit (Agilent). To generate pT7-SA11-NSP3-P2A-VP3-GTase-3xFlag plasmid, a P2A-GTase-3xFlag fragment was inserted into a pT7-SA11-NSP3 plasmid. All the plasmids are purified by using a QIAGEN Plasmid Maxiprep kit per the manufacturer's instructions.

## Antibodies and reagents

Antibodies used in the study included the following: rabbit anti-PFDN3 (A305-403A, Bethyl Laboratories, USA); rabbit anti-PFDN4 (16045-1-AP, Proteintech, China); mouse anti–Flag (F1804, Sigma-Aldrich, USA); mouse anti-Myc (2276S, Cell Signaling Technology, USA); rabbit anti–GAPDH (2118S, Cell Signaling Technology, USA); mouse anti–β actin (4970S, Cell Signaling Technology, USA); rabbit anti-OAS3 (PA5-59539, Invitrogen, USA); rabbit anti-OAS1 (14498S, Cell Signaling Technology, USA); rabbit anti-ISG15 (15981-1-AP, Proteintech, China); rabbit anti-IFITM1 (131263, Cell Signaling Technology, USA); rabbit anti-MX1 (13750-1-AP, Proteintech, China); rabbit anti-IRF3 (4302S, Cell Signaling Technology, USA); rabbit anti-STAT1 (9172S, Cell Signaling Technology, USA); rabbit anti–pSTAT1 (9174S, Cell Signaling Technology, USA); rabbit anti-STAT2 (sc-514193, Santa Cruz Biotechnology, USA); mouse anti-UBA3 (sc-377272, Santa Cruz Biotechnology, USA); rabbit anti–IRF9 (76684, Cell Signaling Technology, USA); horseradish peroxidase–conjugated anti-mouse and anti-rabbit (7076S and 7074S, cell signaling technology, USA); Alexa Fluor 594–conjugated goat anti-mouse (A-11005, Thermo Fisher Scientific, USA); 4′,6′-diamidino-2-phenylindole (1:1000) (00-4959-52, Invitrogen, USA). IFN-β was purchased from R&D Systems (8499-IF/CF). AGI6780 (S7241) and Disulfiram (S1680) were purchased from Selleck Chemicals. MLN4924 was purchased from Sigma-Aldrich (505477). DMSO was purchased from VWR (VWRVN182).

## Tandem affinity purification and mass spectrometry

Stable LAP cell lines were harvested using detergent. Lysates were clarified at 50,000 g and subjected to anti-GFP immunoprecipitation. Bound proteins were eluted from antibody beads using TEV protease, recaptured on S-protein agarose (Millipore), and eluted in 2x NuPAGE sample buffer (Invitrogen). Following purification, great care is taken to ensure a lack of contamination from both environmental sources and from other purified proteins. Each purified set of interacting proteins is separated on an individual 4–12% Bis-Tris polyacrylamide gel and stained with Silver Stain Kit (24612, ThermoFisher Scientific, USA). HEK293 cells samples were run into 20–40 mm gels and divided into 20–40 × 1 mm slices. Each excised lane was reduced, propionamidated, and digested with trypsin. Peptide identification of each digestion mixture was performed by microcapillary reversed-phase HPLC nanoelectrospray tandem mass spectrometry (mLC-MS/MS) on an LTQ-Orbitrap Elite or Fusion mass spectrometer (ThermoFisher Scientific, USA). The Orbitrap repetitively surveyed an m/z range from 395 to 1600, while data-dependent MS/MS spectra on the twenty (Velos) or ten (XL) most abundant ions in each survey scan were acquired in the linear ion trap. MS/MS spectra were acquired with a relative collision energy of 30%, 2.5-Da isolation width, and recurring ions dynamically excluded for 60 s.

Peptide sequencing and protein inference were facilitated using Byonic following an initial quality control analysis using Preview (Protein Metrics, San Carlos, CA). In a typical Byonic analysis, a 12 ppm mass tolerance for precursor ions and 0.4 Da mass tolerance for fragment ions against a species-specific (mouse or human) fasta file derived from the NCBI GenBank protein database with custom sequences added for specific tagged bait protein sequences. Fully specific tryptic peptides were accepted, with up to two missed cleavages, allowing for various common modifications such as methionine oxidation and acetylation of protein N-termini, as well as modifications specific to the pathways investigated. Both protein- and peptide-level false discovery rates were held to an estimated false discovery rate (FDR) of <1% using a reverse decoy database strategy[27].

## TAP/MS data network generation

For individual genes identified in each AP/MS sample, we assigned a normalized spectral abundance factor (NSAF) to each gene[27]. This includes all peptides except those known to derive from the bait protein and those derived from known exogenous proteins. For example, we exclude proteins commonly found in human skin and those added during sample preparation. We then divide this score by the mean length of NCBI reference protein isoforms from g (Lg) in amino acids. Using a set of negative control datasets, we systematically search for genes whose score in an experimental data set is highly unlikely. These filtered genes were included in Supplementary Data 1, and a manually curated, simplified subset of these is shown in Fig. 1a, plotted by Cytoscape v3.9.1.

## Co-immunoprecipitation and western blot

HEK 293 T cells were transfected with indicated plasmids for 48 h; then, the cells were washed with cold phosphate-buffered saline (PBS) and lysed in cell lysis buffer for western blot and immunoprecipitation (FNN0021, Invitrogen, USA) containing protease inhibitor cocktail (04693132001, Roche, Basel, Switzerland). Cell lysates were incubated with antibody and bead at 4 °C overnight. Protein-antibody-bead complexes were washed three times with immunoprecipitation lysis buffer as described above and analyzed by western blot[54]. For GFP immunofluorescence analysis, we used ChromoTek GFP-Trap® Magnetic Agarose which was purchased from Proteintech.

## Immunofluorescence analysis

HEK293 cells were transfected with the indicated plasmids with Lipofectamine 3000 (Invitrogen) according to the manufacturer's protocol. Cells were washed three times with phosphate-buffered saline (PBS) and then fixed with 4% (wt/vol) paraformaldehyde for 15 min at room temperature. Cells were then washed three times with PBS and incubated with 0.1% Triton X-100 for 10 min. Next, 5% bovine serum albumin was used to block for 2 h. Cells were then incubated with the antibody for 2 h. Nuclei were stained with DAPI for 10 min. All cells were washed with PBS 5 times after each step and were imaged by confocal microscopy (Carl Zeiss LSM 880 Confocal Microscope and ZEN 2.3 LITE software)[54,55].

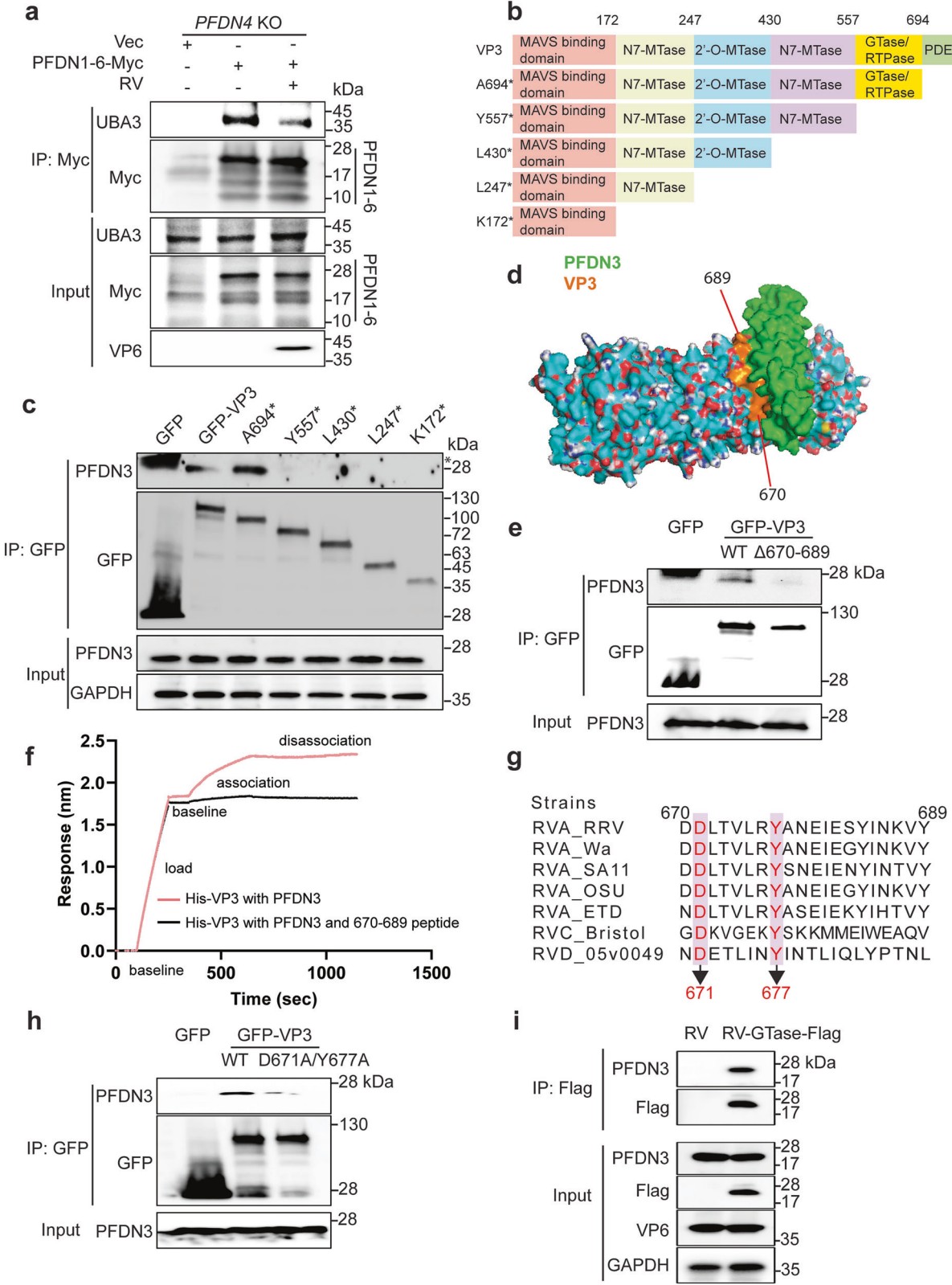

## His pulldown

His-PFDN3 protein was purchased from Abcam (ab139201), and UBA3 protein was purchased from Abcam (ab140421). His-CDPK1 was kindly provided by Dr. David Sibley (Washington University in St. Louis). His-VP3 was used in our previous study[21]. Added 20 µl of cOmplete™ His-Tag Purification Resin (05893801001, Roche, Basel, Switzerland) to 1.5 ml tube and washed with PBS 3 times. Incubate with His-VP3, His-

CDPK1, or His-PFDN3 (a total of 2–5 µg of fusion protein) at 4 °C for 2 h. His-PFDN3 was cleavage the His tag to obtain PFDN3 protein using thrombin, and thrombin was removed by His-Tag purification resin. Washed 3 times with PBS and added the same amount of PFDN3 or UBA3 protein overnight. Centrifuge at 500 x *g* for 5 min and remove the supernatants (keep 20 µl for flow-through detection). Wash the beads with PBS 3 times. Add 2x SDS loading buffer to the beads and

**Fig. 6 | Amino acids 671D and 677Y within VP3 are crucial for hijacking PFDN and inhibiting the interaction between PFDN and UBA3. a** *PFDN4* KO HEK293 cells were transfected with PFDN1-6-Myc plasmids for 24 h and infected with or without RV (RRV strain, MOI of 5) for another 24 h. Cell lysates were subject to immunoprecipitation using α-Myc antibody and probed for α-UBA3 antibody. PFDN3 is the top band, and PFDN1-4 and PFDN6 are similar in size, as shown on the bottom. **b** Schematic diagram of WT and mutant VP3 proteins with defined domains illustrated in colors. **c** HEK293 cells were transfected with GFP-tagged VP3 mutants for 48 h. Cell lysates were subject to immunoprecipitation using α-GFP antibody and probed for α-PFDN3 antibody. * represents a non-specific band. **d** A predicted structure of VP3 and PFDN3 complex by AlphaFold-3 (PFDN3: green, VP3: cyan). The orange area was the region 670-689 that interacts with PFDN3. Surface electrostatic charge includes red, positive charge; blue, negative charge; and white, neutral charge. **e** HEK293 cells were transfected with indicated plasmids for 48 h.

Cell lysates were subject to immunoprecipitation using α-GFP beads and probed for α-PFDN3 antibody. **f** The PFDN3 protein was incubated with 670-689 peptide. BLI sensorgrams obtained using biosensor loaded with His-tagged VP3 (10 nM) (load) and incubated with whether 50 μM PFDN3 protein or PFDN3-670-689 peptide (association). The dissociation is shown on the right. **g** Amino acid alignment of VP3 (670-689) between different RV strains. Group A rotavirus (RVA: RRV, simian strain; Wa, human strain; SA11, simian strain; OSU, porcine strain; ETD, murine strain), Group C rotavirus (RVC: Bristol, human strain), Group D rotavirus (RVD: 05v0049, chicken strain). **h** HEK293 cells were transfected with GFP vector, GFP-VP3, or GFP-VP3-D671A/Y677A plasmids for 48 h. Cell lysates were subject to immunoprecipitation using α-GFP beads and probed for α-PFDN3 antibody. **i** HT-29 cells were infected with SA11 or SA11-GTase-Flag (MOI = 0.1) for 24 h. Cell lysates were subject to immunoprecipitation using α-Flag antibody and probed with α-PFDN3 antibody. For **a**, **c**, **e**, and **h–i** were repeated at least two times.

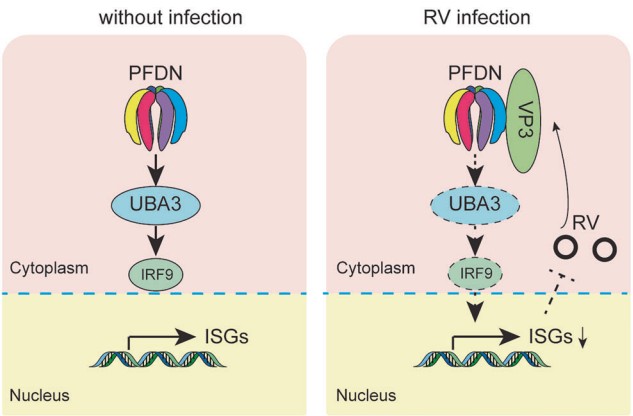
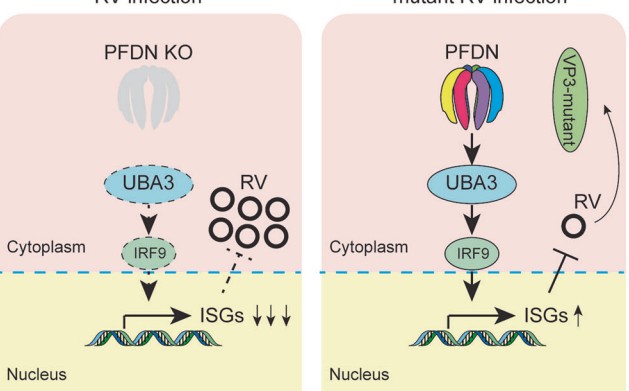

**Fig. 7 | A working model of PFDN-RV VP3 interactions.** PFDN interacts with UBA3 and ensures its correct folding. UBA3 is essential for ISG production. In RV-infected cells, VP3 protein competes with UBA3 for interaction with PFDN. This competition results in the inhibition of ISG expression, thereby benefiting RV replication. In PFDN deficient cells, UBA3 levels are decreased, leading to inhibited ISG production and increased virus replication. RV with VP3 critical mutations cannot bind PFDN and inhibit UBA3 and thus have reduced infectivity. Image was created using Adobe Illustrator.

then boil for 10 min. Analysis of fusion proteins by using 10 μl aliquots of each sample to run the SDS-gel. Stain with Coomassie brilliant blue to visualize the pulldown protein. The target proteins in the eluates were detected by western blot.

### Bio-layer interferometry (BLI) assays

The binding profile and apparent binding affinity of VP3 and PFDN3, and PFDN3 and UBA3 were measured by a BLI using Octet-Red96 instrument (ForteBio). For some experiments, His-tagged VP3 or His-tagged CDPK1 were loaded onto Ni-NTA biosensors. Alternatively, non-Tagged PFDN3 or GST-tagged UBA3 were incubated with these proteins. The VP3 peptide was synthesized by Gencript. The VP3 peptide sequence is as follows: DDLTVLRYANEIESYINKVY. Experiments were carried out in PBS at 30 °C. Data were analyzed, and the binding curves were fit using the Data Analysis 9.0 software package (ForteBio). Association and dissociation curves were exported using Excel format and imported into Prism (GraphPad software) for visualization of the sensorgram.

### CRISPR/Cas9 knockout cells

Single clonal knockout HEK293 cells were obtained using the PX458 vector that expresses Cas9 and sgRNAs against PFDN3 (Supplementary Table 1). GFP-positive single cells were sorted at 48 h post-transfection using BD Aria II into 96-well plates and screened for knockout based on Sanger sequencing. Pooled UBA3 knockout HEK293 cells were obtained by lentiviral transduction with the lenti-CRISPR_v2, psPAX2, and pMD2.G vector that expresses Cas9 and sgRNA for a minimum of

14 days under puromycin selection. sgRNAs against UBA3. The *PFDN3* and *PFDN4* genes were amplified with primers (Supplementary Table 1). Then the DNA was purified with a kit and sent for Sanger sequencing.

### RT-qPCR

The total RNA of the HEK293 cells (WT, *PFDN* KO, *UBA3* KO) that were infected with recombinant UK, WI61, and OSU virus, or stimulated with or without IFN-β or treated with inhibitors was extracted by RNeasy Mini Kit (74106, Qiagen, USA). Total RNA was reverse transcribed to cDNA using a high-capacity cDNA reverse transcription kit (4368814, ThermoFisher Scientific, USA) with RNase inhibitor (N8080119, ThermoFisher Scientific, USA) according to the user guide. The expression level of Reovirus, host gene IFITM1, MX1, OAS3, OAS1, ISG15, and housekeeping gene GAPDH was quantitated by 2× SYBR green master mix (4368708, ThermoFisher Scientific, USA), and NSP5 was quantitated by 2× TaqMan Fast Advanced master mix (4444557, ThermoFisher Scientific, USA). The primers used in this study are shown in Supplementary Table 1. The y axis stands for the percentage of NSP5 mRNA levels relative to GAPDH levels[14].

### RNase L-mediated rRNA cleavage assay

WT and *PFDN4* KO HEK293 cells were transfected with 1 μg/ml of poly(I·C) or 2−5 A for 6 h. Total RNA was extracted using a Thermo-Fisher PureLink RNA mini kit and resolved by an RNA chip assay. The cleavage products of 18S and 28S rRNAs were quantitated using the Bioanalyzer system as described previously[25]. For quantification, the

areas below the curves of the regions corresponding to 18S and 28S and those between 18S and 28S were calculated from the electropherogram, and their sum was taken as 100%. The fraction represented by each band was then calculated[26].

## Focus-forming assays

Activated virus samples from cell culture were serially diluted 10-fold and added to confluent monolayers of MA104 cells seeded in 96-well plates for 1 h at 37 °C. Inoculates were removed and replaced with M199 serum-free and then incubated for 16 to 18 h at 37 °C. Cells were then fixed with 10% paraformaldehyde for 30 min and permeabilized with 1% Triton for 3 min. Cells were incubated with rabbit hyperimmune serum to RRV strain produced in our laboratory and previously described[56] and anti-rabbit HRP-linked secondary antibody. Viral foci were stained with 3-amino-9-ethylcarbazole (AEC substrate kit. Vector Laboratories) per the manufacturer's instructions and enumerated visually.

## Flow cytometry analyses

HEK293 cells were infected with MOI of 0.01 VSV-GFP virus for 24 h. Then cells were digested with trypsin and harvested for flow analysis, which was described previously[55].

## RNA-seq

Total RNA was extracted from WT, PFDN4 KO, and PFDN4 rescue HEK293 cells. RNA-seq libraries were constructed using TruSeq RNA single-index adapters and deep sequenced as described below at Washington University in St. Louis, MO. Briefly, total RNA integrity was determined using Agilent Bioanalyzer or 4200 Tapestation. Library preparation was performed with 5 μg of total RNA with a Bioanalyzer RIN score greater than 8.0. Ribosomal RNA was removed by poly-A selection using Oligo-dT beads (mRNA Direct kit, Life Technologies). mRNA was then fragmented in reverse transcriptase buffer and heated to 94 degrees for 8 min. mRNA was reverse transcribed to yield cDNA using SuperScript III RT enzyme (Life Technologies, per manufacturer's instructions) and random hexamers. A second strand reaction was performed to yield ds-cDNA. cDNA was blunt-ended, had an A base added to the 3' ends, and then had Illumina sequencing adapters ligated to the ends. Fragments were sequenced on an Illumina NovaSeq-6000 using paired-end reads extending 150 bases. RNA-seq reads were then aligned and quantitated to the Ensembl release 101 primary assembly with an Illumina DRAGEN Bio-IT on-premise server running version 3.9.3-8 software. To find the most critical genes, the Limma voomWithQualityWeights transformed $\log_2$ counts-per-million expression data was then analyzed via weighted gene correlation network analysis with the R/Bioconductor package WGCNA.

## Label-free quantification (LFQ) mass spectrometry

WT, PFDN3 knockout, and PFDN4 knockout HEK293 cells were stimulated with 500 U/ml IFN-b for 24 h. The cell pellets were lysed in the lysis buffer, and the supernatants containing proteins were used for the BCA assay to determine protein concentration. For each sample, 500 μg of proteins were purified by acetone/TCA precipitation. Samples were reduced with 4 mM dithiothreitol at 50 °C for 30 min, and cysteines were alkylated with 18 mM iodoacetamide at room temperature for 30 min in the dark. The solution was diluted to <2 M urea and trypsin was added with the 1:50 ratio (trypsin: protein) at 37 °C for overnight incubation. The resulting peptides were desalted using solid-phase extraction on a C18 Spin column and eluted in 80% acetonitrile in 0.1% formic acid. Peptides were injected into a Neo trap cartridge coupled with an analytical column (75 μm ID x 50 cm PepMapTM Neo C18, 2 μm). Samples were separated using a linear gradient of solvent A (0.1% formic acid in water) and solvent B (0.1% formic acid in ACN) over 120 min using a Vanquish Neo UHPLC

System coupled to an Orbitrap Eclipse Tribrid Mass Spectrometer (ThermoFisher Scientific). Protein tandem MS data was queried for protein identification and label-free quantification against the SwissProt human database using MaxQuant. The following modifications were set as search parameters: trypsin digestion cleavage after K or R (except when followed by P), 2 allowed missed cleavage site, carbamidomethylated cysteine (static modification), and oxidized methionine, protein N-term acetylation (variable modification). Search results were validated with peptide and protein FDR both at 0.01. Proteins that were identified with >1 unique peptide were subjected to a t-test using Perseus software. Mass Spectrometry analyses were performed by the Mass Spectrometry Technology Access Center at the McDonnell Genome Institute (MTAC@MGI) at Washington University School of Medicine. The mass spectrometry proteomics data have been deposited to the ProteomeXchange Consortium via the PRIDE partner repository with the dataset identifier PXD056198.

## Generation of recombinant RVs

rSA11-RRV-VP3, rSA11-RRV-VP3-D671A/Y677A, and rSA11-GTase-Flag viruses were generated using the following pT7 plasmids: pT7-SA11-VP1, -VP2, -VP4, -VP6, -VP7, -NSP1, -NSP2, -NSP3, -NSP4, and -NSP5 according to the optimized entirely plasmid-based RG system[51]. The pT7-SA11-VP3 plasmid was replaced by the pT7-RRV-VP3, pT7-VP3-D671A/Y677A and pT7-SA11-NSP3-VP3-GTase-Flag to generate rSA11-RRV-VP3, rSA11-RRV-VP3-D671A/Y677A, and rSA11-GTase-Flag. We cotransfected BHK-T7 cells with T7 plasmids together with C3P3-G3 plasmid and overlayed with *SERPINB1* knockout MA104 cells to generate the viruses[42]. The rescued recombinant RVs were propagated for two passages in MA104 cells in a 6-well plate and then were plaque-purified twice in MA104 cells.

## In vivo experiments

C57BL/6 mice were purchased from the Jackson Laboratory and bred locally at the Washington University in St. Louis (WUSTL) CSRB vivarium. Five-day-old suckling pups were orally infected with SA11 and SA11-VP3-D671A/Y677A viruses at an inoculum of $1.5 \times 10^3$ FFU. At 24 h post infection, intestinal tissues were collected for RNA extraction and subsequent RT-qPCR analysis. We have not observed any sex-dependent phenotypes in rotavirus susceptibility in vivo, therefore we used both male and female mice in the current study.

## Ethics statement

All animal studies were approved by Washington University in St. Louis Institutional Animal Care and Use Committee (IACUC) with the protocol number 22-0269. Mice were maintained in a temperature- and humidity-controlled BSL-2 barrier facility with veterinary oversight and unrestricted access to food and water. Challenge studies with selected agents were performed under approved protocols by trained personnel in a certified BSL-2 facility. Adult mice were euthanized by $CO_2$ inhalation followed by cervical dislocation, while neonatal pups were euthanized by decapitation, in accordance with American Veterinary Medical Association guidelines.

## Statistical analysis

The values in the figures are presented as means ± SE. Student's t-test was used for two-group comparisons to examine statistical significance. ANOVA was used for multiple-group comparisons, followed by Tukey's, Šídák's, or Dunnett's multiple comparisons tests to examine statistical significance. The number of independent experiments performed is indicated in the relevant figure legends.

## Reporting summary

Further information on research design is available in the Nature Portfolio Reporting Summary linked to this article.

## Data availability

RNA-seq raw data, please visit the Sequence Read Archive (SRA), accession number: PRJNA1164811. LFQ raw data are available via ProteomeXchange with the identifier PXD056198. All data needed to evaluate the conclusions in the paper are present in the paper and/or the Supplementary Materials. Source data are provided with this paper.

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

## Acknowledgements

This study is supported by the National Institutes of Health (NIH) grants R01 AI188179 and U19 AI116484 to S.D., and R01 AI036040 to B.V.V.P. We extend our gratitude to the members of the Ding lab for their insightful discussions on this project. We would like to thank Igi Vilza from the Sibley lab at Washington University in St. Louis for providing the His-CDPK1 protein. We are also grateful to Dr. Zhou Huang from the Sibley lab at Washington University in St. Louis for assistance with the BLI assay and to Dr. Sean Whelan (Washington University in St. Louis) for providing the BLI machine. Mass Spectrometry analyses were performed by the Mass Spectrometry Technology Access Center at the McDonnell Genome Institute (MTAC@MGI) at Washington University School of Medicine, supported by the Diabetes Research Center/NIH grant P30 DK020579, Institute of Clinical and Translational Sciences/NCATS CTSA award UL1 TR002345, and Siteman Cancer Center/NCI CCSG grant P30 CA091842.

## Author contributions

Conceptualization: Y.Z. and S.D. Methodology: Y.Z., D.K., and P.K.J. Investigation: Y.Z. and Y.S. Visualization: Y.Z. Supervision: S.D. and B.V.V.P. Writing-original draft: Y.Z. Writing-review and editing: Y.Z., S.D., and B.V.V.P.

## Competing interests

The authors declare no competing interests.
