## [Transparent Peer Review file · Nature Communications]

Prefoldin complex promotes interferon-stimulated gene expression and is inhibited by rotavirus VP3

Corresponding Author: Dr Siyuan Ding

Version 0:

Reviewer comments:

Reviewer #1

(Remarks to the Author)

Review summary:

In this paper, Zhu et al. reveal a novel way in which the rotavirus VP3 protein impacts the host innate immune response. They demonstrate that VP3 binds the prefoldin complex, a protein complex involved in protein folding. Knockout of members of the prefoldin complex such as PFDN3 result in enhanced replication of rotavirus. In absence of prefoldin complex members, expression of interferon stimulated genes (ISGs) is diminished. According to the authors, this occurs because VP3 bound prefoldin complex fails to stabilize UBA3, a critical substrate of prefoldin. UBA3 is required for controlling the levels of IRF9 and consequently the expression of ISGs. The authors identify residues in VP3 that are important for interaction with prefoldin. They also demonstrate that rotavirus mutants with changes to these residues induce greater expression of ISGs and display diminished replication.

Overall, the data presented on this manuscript are novel and fascinating. However, I believe that additional experiments are needed to support the conclusions reached.

1. The interaction of VP3 with prefoldin proteins is demonstrated in transfected cells. Evidence that this interaction occurs in infected cells is important given that VP3 isn't abundantly expressed in infected cells. Transfected cells are likely to have a significantly higher level of VP3 protein so whether this is a relevant pathway in rotavirus infected cells is unclear. A previous study, PMC10746195, has been able to successfully demonstrate interaction between VP3 and cellular proteins in infected cells. Given that the investigators are able to generate mutant rotaviruses by reverse genetics, this should be possible.

Relatedly, I am not sure the immunofluorescence adds anything to the results. Both proteins of interest are overexpressed distributed throughout the cytoplasm. Due to this, their true subcellular localization is unknown. Further, if they are diffusely cytoplasmic, their "colocalization" in the cytoplasm doesn't necessarily help strengthen the argument that they interact.

2. The authors use MLN2924 to suggest that UBA3 may be involved. While it is clear from subsequent experiments that UBA3 interacts with PFDN3 and that this interaction is important for controlling innate immunity, the MLN data showed in Figure 4 do not necessarily support this idea. Ubiquitination plays a critical role in IFN signaling pathways. All such pathways are likely affected by treatment with MLN2924 since it is likely to inhibit most cullin ubiquitin ligases.

3. The authors demonstrate that VP3 inhibits interaction between PFDN3 and UBA3. While the majority of the data are suggestive of this conclusion, measuring dissociation of PFDN3 and UBA3 with increasing doses of VP3 would strengthen this argument. Further, an important piece of the story is whether VP3 is a client protein of prefoldin. This idea isn't explored.

4. The authors demonstrate virus replication is enhanced in absence of prefoldin complex protein. A similar experiment is needed for UBA3 deficient cells. Based on the authors' model, UBA3 deficiency should also lead to enhanced replication. It is suggested that this enhanced replication is due to a diminished innate immune responses. This connection in this system is not established. To do so, the authors would need to perform these experiments in control and knockout cells under conditions where IFN signaling is blocked.

5. The authors demonstrate that VP3 mutant viruses with changes in domain that are predicted to interact with prefoldin are impaired for replication in intestinal enteroids. If the model proposed by the authors' is correct, these viruses should also be impaired for replication in HEK293 cells used in other parts of the manuscript. Further, difference in replication of these viruses in HEK293 should be normalized in absence of PFDN3 or UBA3. Since VP3 is a multifunctional protein, in absence

of these results, it cannot be ruled out if the effect of VP3 mutations in the putative prefoldin binding region affect other functions of VP3 and thereby impair replication in enteroids.

Reviewer #2

(Remarks to the Author)

Zhu et al describe the identification of a novel interaction between rotavirus VP3 and the prefoldin complex. This work aims to address the open questions about how VP3 can help to control immune response in rotavirus infected cells. These are very important and timely questions; however the lack of experiment using virus infections, the lack of relevant cells models and several controls are critically missing to determine the authors claims. Please see below for specific comments:

Major Points:

1. Overexpressing VP3 in HEK cells does not show real interactions in the context of infection. Need to show that VP3 interacts with PFDN3 when other rotavirus proteins are around. The authors mention that VP3 also interacts with MAVS and OAS. However, these proteins are missing from their pull-down (Fig. 1A). Does this mean that VP3 only interacts with these proteins during infection and not when expressed alone? The lack of known interactors being found with VP3 cause doubts that the HEK293 over expression system is mimicking interactions that occur during infection.
2. In addition, HEK293T cells often lack many factors required to stimulate and induce ISGs. Would be important to show that interactions between VP3 and PFDN3/4/5 occur in more physiologically relevant cells with known intact immune sensing pathways.
3. Confocal images in 1C and Sup 2 do not show any evidence of co-localization. The magnification is too low and they are both cytoplasmic which does not show that co-localization is occurring.
4. The authors show that PFDN3 shows the greatest interaction with VP3. PFDN3 KO cells were made however they were left out of many experiments. It is unclear why they were not used to test bovine RV in Fig. 2C. It was also not clear why they were excluded from the RNA-Seq analysis in Fig 3A, OAS3 analysis in Fig 3B, and MX1 analysis in Fig. 3D.
5. The observation that PFDN3 cells also lose PFDN4 is very concerning. This makes any data shown with the PFDN3 KO cells hard to interpret. The authors need to show all PFDN components in their PFDN3 and PFDN4 knock-down cells to clarify this matter. Also, it would be important to show multiple clones of these KO cells. The authors have shown this for their UBA3 KO cells but did not do this for PFDN3 and PFDN4.
 - a. Can the authors re-overexpress PFDN4 in their PFDN3 KO cells to see if loss of PFDN3 alone has a phenotype. In the current version of the manuscript, it is difficult to make any conclusions whether PFDN3 is playing a role or whether all phenotypes are driven by PFDN4 exclusively.
6. From Figure 1B, PFDN3 and PFDN5 were the proteins that were shown to interact the most with RV Vp3 so it is unclear why PFDN5 was also not analyzed by KO and how PFDN4 was chosen.
7. Figure 4E is missing a control to show that another myc-tagged protein would also not interact with UBA3.
8. In Figure 5C, the authors show that PFDN4 KO cells also have low levels of UBA3. How can the authors separate out these functions. From the current state, it seems like loss of UBA3 is the dominant phenotype and not loss of PFDN4.
9. Figure 6 missing control to show the amount of RV infection and VP3 expression.

Minor Points:

1. In 6B it's not clear that these are truncated mutants and not point mutants. A better schematic is needed. Also, in this figure it seems like GFP alone is associating with PFDN3.
2. No sure what MA104 SERPINB1 cells were used for – they were listed in Materials and Methods.

Reviewer #3

(Remarks to the Author)

The manuscript "Prefoldin complex promotes interferon-stimulated gene expression and is inhibited by rotavirus VP3" by Zhu et al. reports a novel prefoldin-UBA3-IRF9-ISG axis in antiviral innate immunity, and VP3 interfered this axis through direct and competitive binding to PFDNs. The authors identified PFDNs as a key protein family that interacts with VP3 by protein-protein interaction identification and reverse genetics, and revealed a novel role for PFDNs in antiviral immunity, which is of great significance in expanding not only the understanding of rotavirus VP3 antagonism to interferon signaling, but also new mechanisms in the field of antiviral resistance. However, there are still some revisions need to be clarified and the comments are as follows:

Major:

1. The authors mainly applied PFDN3 and PFDN4 from PFDNs to study, some experiments applied both PFDN3 and PFDN4, some only applied one of them, what is the reasons for this? For example, according to the manuscript, VP3 directly interacts with PFDN3 and knockout of PFDN3 would simultaneously make PFDN4 almost undetectable, so it seems that PFDN3 is more significant, whereas the initial characterization of the role of PFDN in the RV (Fig.2C) and the effect on ISG (Fig.3A) just applied PFDN4 but not PFDN3, what is the reason for this?

2. The VP3 protein used for interactions in the text is from RRV, and when verifying the effect of PFDN on RV, UK, WI61, and OSU strains were applied, but not RRV, what is the reason for this?
3. In the paper, there are only results of the effect on RV after PFDN knockout, is it possible to add some PFDN overexpression data?
4. As in Fig. 2C, at 24 h, PFDN seems to be proviral (although not statistically significant), so would PFDN have a role in promoting rotavirus infection at earlier time points? Because PFDN affect actin polymerization, which may participate in RV entry.
5. The text identifies ISGs expression mainly using MX1 as the indicator, and the main reason for choosing this indicator and related references should be added to the text.
6. For PFDNs and UBA3, it is suggested that UBA3 interacted with mix of PFDNs plasmids (Fig. 4E), and then demonstrated that UBA3 directly interacted with PFDN3. I wonder if any attempts were made to characterized the direct interactions of UBA3 with other PFDNs, which may affect the ability of VP3 to block the interactions of PFDNs and UBA3. Additionally, in the further experiments to characterize the role of UBA3 in ISG production, it was switched to PFDN4 KO cells (Fig. 5), and the reason for this should be explained clearly in the text.
7. What's relationship of PFDN3 and PFDN4, complementary, independent, or mutually enhancing? Did the author perform double KO experiment of PFDN3 and PFDN4.
8. Additionally, what's exact function of PFDNs to UBA3? assists in folding (line 23), PFDN operates upstream of UBA3 (line 235), or PFDN stabilizes UBA3 (line 236)? These are different biological processes. And it's related to the Fig. 8, the first figure of the working model.
9. In introduction, the authors mentioned that PFDN seems to be proviral in previous publications. But our results proved that PFDN possesses antiviral function? Why does this difference exist in different viruses? The authors should discuss this in the discussion part.
10. The in vivo experiment are important to the conclusion.

Other comments

1. Some words should be in its full name because it's appeared in the first time, such as RVs, MAVS, etc.. Please double check.
2. For PFDN3, its current gene symbol is VBP1. Of course, there is not a problem with calling it PFDN3 to show that it is part of PFDNs, but it's better to explain in the text so as not to make confuse.
3. Line 41: The reference here is maybe wrong, and it looks like the following:
Mousnier A, Kubat N, Massias-Simon A, Ségéral E, Rain JC, Benarous R, Emiliani S, Dargemont C. von Hippel Lindau binding protein 1-mediated degradation of integrase affects HIV-1 gene expression at a postintegration step. *Proc Natl Acad Sci U S A*. 2007 Aug 21;104(34):13615-20. doi: 10.1073/pnas.0705162104. Epub 2007 Aug 13. PMID: 17698809
4. Line 125: The figure here should be (Supplementary Fig. 7B).
5. Line 150-152: "...we treated cells with small-molecule inhibitors AGI6780 (33), MLN4924 (34), and Disulfiram (35), which block UBA3, IDH2, and ALDH2, respectively. Among the compounds tested, only MLN4924, an inhibitor of UBA3 and the neddylation pathway..." There is a contradiction of the sentence. In the first part, AGI6780 is inhibitors of UBA3; in the later part, it seems MLN4924 is inhibitor of UBA3. Please double check.
6. Line 154: Why do you think 100 nM is very low concentration? How about 1 or 10 nM?
7. Line 173-174: UBA3 protein level was almost undetectable in PFDN4 knockout cells. Why did this happen? The author need to be explained somewhere.
8. Line 233-234: This sentence is a bit exaggerated, "We have demonstrated that knocking out individual PFDN subunits significantly enhances viral replication". The authors only studied PFDN3 and 4, not each individual PFDN subunits.
9. Line 256: "... IRF9 through weather through neddylation modification..." is incorrect.
10. Line 264-271: The section is unnecessary.
11. Line 285: I couldn't find the cell lines in the results, "MA104 SERPINB1 knockout cells"
12. Line 287: The information of these RV strains need to be provided.
13. Line 439: The full stop should be deleted.

Reviewer #4

(Remarks to the Author)

Version 1:

Reviewer comments:

Reviewer #1

(Remarks to the Author)

The authors have done a commendable job and have responded to the majority of comments in a satisfactory manner.

However, one response remains unsatisfactory: the explanation regarding the replication phenotype of the mutant viruses. It is unclear whether this phenotype is due to VP3's ability to counteract the prefoldin complex or due to other functions of VP3. Although the authors have generated tools (e.g., knockout cells) to address this question directly, they have not utilized them. Furthermore, because viral replication during the early time points (1–8 hours) is minimal, I do not believe the data

support a definitive conclusion about which step of infection is affected by the VP3 mutation.

Reviewer #2

(Remarks to the Author)

The authors have addressed all comments from the first revision. My only concern is still lack of the infection model. They use an infection with over expression to show the interaction in this revised version. However, this is still a bit concerning as there is still so much protein being expressed that it may push interactions that are not normally present under real infection conditions.

Reviewer #3

(Remarks to the Author)

The authors have responded all my comments and concerns.

One more suggestion, it's better to integrate all the sequence of sgRNAs or primers into one table.

Reviewer #4

(Remarks to the Author)

Response to the reviewers' comments on *Nature Communications* manuscript (ID: NCOMMS-24-84308, Zhu Y et al., Prefoldin complex promotes interferon-stimulated gene expression and is inhibited by rotavirus VP3).

Below please find our point-by-point response to the reviewers' comments:

Reviewer #1:

Review summary:

In this paper, Zhu et al. reveal a novel way in which the rotavirus VP3 protein impacts the host innate immune response. They demonstrate that VP3 binds the prefoldin complex, a protein complex involved in protein folding. Knockout of members of the prefoldin complex such as PFDN3 result in enhanced replication of rotavirus. In absence of prefoldin complex members, expression of interferon stimulated genes (ISGs) is diminished. According to the authors, this occurs because VP3 bound prefoldin complex fails to stabilize UBA3, a critical substrate of prefoldin. UBA3 is required for controlling the levels of IRF9 and consequently the expression of ISGs. The authors identify residues in VP3 that are important for interaction with prefoldin. They also demonstrate that rotavirus mutants with changes to these residues induce greater expression of ISGs and display diminished replication.

Overall, the data presented on this manuscript are novel and fascinating. However, I believe that additional experiments are needed to support the conclusions reached.

A: We thank the reviewer for the positive summary of our work. We have performed additional experiments to address the raised questions, including VP3 and PFDN3 interaction under infection conditions, competition between UBA3 and VP3 for PFDN, rotavirus (RV) replication in UBA3 knockout (KO) cells, interferon (IFN) signaling blocking assays in UBA3 KO cells, and VP3 expression in PFDN knockout cells. We have modified the manuscript carefully based on these new additions. Please see the new Fig. 1d, 5e, 6i, 7b, Supplementary Fig. 5, Supplementary Fig. 13, and Revision Fig. 1.

Q1. The interaction of VP3 with prefoldin proteins is demonstrated in transfected cells. Evidence that this interaction occurs in infected cells is important given that VP3 isn't abundantly expressed in infected cells. Transfected cells are likely to have a significantly higher level of VP3 protein so whether this is a relevant pathway in rotavirus infected cells is unclear. A previous study, PMC10746195, has been able to successfully demonstrate interaction between VP3 and cellular proteins in infected cells. Given that the investigators are able to generate mutant rotaviruses by reverse genetics, this should be possible.

Relatedly, I am not sure the immunofluorescence adds anything to the results. Both proteins of interest are overexpressed and distributed throughout the cytoplasm. Due to this, their true subcellular localization is unknown. Further, if they are diffusely cytoplasmic, their “colocalization” in the cytoplasm doesn’t necessarily help strengthen the argument that they interact.

A1. We thank the reviewer for the helpful comments. To validate the VP3-PFDN3 interaction under infection conditions, we performed two new experiments. (1) We transfected GFP-VP3 into A549 cells and then super-infected them with rotavirus. Co-immunoprecipitation confirmed PFDN3 interaction with VP3 but not the GFP empty vector in an infected cell type that is interferon-competent (new Fig. 1d; lines 79-80).

New Fig. 1d. VP3 interacts with PFDN3 in rotavirus-infected A549 cells.

A549 cells were transfected with the indicated plasmids for 24 hours, followed by infection with rotavirus at an MOI of 1 for 12 hours. Co-immunoprecipitation was performed, and the interaction between VP3 and PFDN3 was analyzed by western blot.

(2) Although the effort to rescue a recombinant rotavirus expressing epitope tagged full-length VP3 (PMID: 37905816) was unsuccessful, despite multiple attempts, we were able to rescue a recombinant rotavirus expressing a Flag-tagged VP3 GTase/RTPase domain (RV-GTase-Flag). The recombinant virus replicated comparably to the parental RV in MA104 cells, and the modified NSP3-P2A-GTase-Flag gene was confirmed by sequencing (new Supplementary Fig. 13a and b). Co-immunoprecipitation in infected HT-29 cells confirmed that the GTase domain interacted with endogenous PFDN3 (new Fig. 6i; lines 221–228).

New Fig. 6i. VP3-GTase domain interacts with PFDN3 during RV infection.

HT-29 cells were infected with wild-type or SA11 expressing VP3-GTase-Flag (MOI=0.1) for 24 hours. Cell lysates were subject to immunoprecipitation using α -Flag antibody and probed with α -PFDN3 antibody.

New Supplementary Fig. 13. Characterization of an RV expressing Flag-tagged VP3-GTase domain.

(a) MA104 cells were infected with either parental RV or RV-VP3-GTase-Flag virus at an MOI of 0.01. Virus titers were determined by an FFU assay at 1, 8, 24, and 48 hours post infection.

(b) The NSP3-P2A-VP3-GTase-Flag gene was amplified from virus stock and verified by Sanger sequencing.

We also fully agree with the reviewer's comment regarding the limited value of the immunofluorescence data. As both VP3 and PFDN proteins were overexpressed and displayed diffuse cytoplasmic distribution, the observed colocalization does not provide additional support for the interaction. Therefore, we have removed the immunofluorescence panel from the revised manuscript.

Q2. The authors use MLN2924 to suggest that UBA3 may be involved. While it is clear from subsequent experiments that UBA3 interacts with PFDN3 and that this interaction is important for controlling innate immunity, the MLN data showed in Figure 4 do not necessarily support this idea. Ubiquitination plays a critical role in IFN signaling pathways. All such pathways are likely affected by treatment with MLN2924 since it is likely to inhibit most cullin ubiquitin ligases.

A2. We thank the reviewer for this important point. We agree that MLN4924, despite being a specific neddylation inhibitor, likely affects multiple cullin-RING ligases involved in IFN signaling, and therefore, the results from MLN4924 treatment alone do not specifically implicate UBA3. To address this, we complemented the MLN4924 data with experiments using *UBA3* KO cells (Fig. 5b-d), which more directly demonstrate that UBA3, the E1 ligase of the neddylation pathway, is required for the proper induction of interferon-stimulated genes (ISGs). We have added new data showing enhanced rotavirus infection in *UBA3* KO cells (new Fig. 5e). While the MLN4924 data suggest that the neddylation pathway as a whole contributes to ISG regulation, our genetic evidence using *UBA3* KO cells highlights a more specific role for UBA3 in this process.

New Fig. 5e. Ruxolitinib does not enhance rotavirus replication in *UBA3* KO cells.

Wild-type (WT) and *UBA3* KO HEK293 cells were treated with or without Ruxo (10 μ M) and IFN- β (500U/ml) for 24 h, and then infected with UK (MOI=5) for 72 h. Virus titers were determined by an FFU assay. Data represent the average of three experiments; error bars indicate SEM (one-way ANOVA with Tukey's post hoc test; *, P<0.05; ****, P<0.0001; ns, not significant).

Q3. *The authors demonstrate that VP3 inhibits interaction between PFDN3 and UBA3. While the majority of the data are suggestive of this conclusion, measuring dissociation of PFDN3 and UBA3 with increasing doses of VP3 would strengthen this argument. Further, an important piece of the story is whether VP3 is a client protein of prefoldin. This idea isn't explored.*

A3. **We thank the reviewer for these insightful suggestions. Since the VP3 expression from the LAP6 construct is generally low, we conducted the reciprocal experiment using increasing doses of UBA3. Our results indicate that UBA3 inhibits the interaction between VP3 and PFDN3 in a dose-dependent manner (Revision Figure 1).**

Revision Fig. 1. Increasing UBA3 expression inhibits the interaction between VP3 and PFDN3.

HEK293 cells were co-transfected with GFP-VP3 and increasing amounts of UBA3 expression plasmid for 48 hours. Cell lysates were subjected to immunoprecipitation and western blot analysis.

In addition, we investigated whether VP3 is a client protein of the PFDN complex by analyzing VP3 protein levels in *PFDN3* and *PFDN4* KO cells using immunofluorescence (New Supplementary Fig. 5a) and flow cytometry (New Supplementary Fig. 5b, c). We observed no significant change in VP3 levels compared to WT cells, suggesting that VP3 is unlikely to be a direct substrate of prefoldin. This is also supported by the observation of enhanced instead of reduced RV infection in *PFDN* KO cells (Fig. 2). Please refer to the new Supplementary Fig. 5, lines 98-99.

New Supplementary Fig. 5. Knockout of *PFDN3* and *PFDN4* does not affect VP3 protein expression.

WT, *PFDN3* KO, and *PFDN4* KO HEK293 cells were transfected with LAP6-VP3 plasmid for 48 hours. VP3 expression was analyzed by immunofluorescent microscopy (a) and flow cytometry (b and c).

Q4. The authors demonstrate virus replication is enhanced in absence of prefoldin complex protein. A similar experiment is needed for *UBA3* deficient cells. Based on the authors' model, *UBA3* deficiency should also lead to enhanced replication. It is suggested

that this enhanced replication is due to a diminished innate immune responses. This connection in this system is not established. To do so, the authors would need to perform these experiments in control and knockout cells under conditions where IFN signaling is blocked.

A4. We thank the reviewer for this important suggestion. To test whether UBA3 deficiency enhances virus replication, we measured virus replication in UBA3 KO cells and observed increased viral replication compared to WT controls (new Fig. 5e).

To investigate whether this effect is due to a diminished innate immune response, we treated WT and UBA3 KO cells with ruxolitinib, a pan-JAK inhibitor, to block STAT1 signaling downstream of IFN- β treatment. Ruxolitinib treatment enhanced virus replication in WT cells but had no additional effect in UBA3 KO cells. These results support the conclusion that enhanced viral replication in UBA3-deficient cells is due to impaired IFN signaling. Please refer to the new Fig. 5e, lines 190-193.

New Fig. 5e. Ruxolitinib does not enhance rotavirus replication in UBA3 KO cells.

Wild-type (WT) and UBA3 KO HEK293 cells were treated with or without Ruxo (10 μ M) and IFN- β (500U/ml) for 24 h, and then infected with UK (MOI=5) for 72 h. Virus titers were determined by an FFU assay. Data represent the average of three experiments; error bars indicate SEM (one-way ANOVA with Tukey's post hoc test; *, P<0.05; ****, P<0.0001; ns, not significant).

Q5. The authors demonstrate that VP3 mutant viruses with changes in domain that are predicted to interact with prefoldin are impaired for replication in intestinal enteroids. If the

model proposed by the authors' is correct, these viruses should also be impaired for replication in HEK293 cells used in other parts of the manuscript. Further, difference in replication of these viruses in HEK293 should be normalized in absence of PFDN3 or UBA3. Since VP3 is a multifunctional protein, in absence of these results, it cannot be ruled out if the effect of VP3 mutations in the putative prefoldin binding region affect other functions of VP3 and thereby impair replication in enteroids.

A5. We thank the reviewer for this insightful comment. We agree that it is critical to determine whether the observed replication defects of the VP3 mutant viruses are due specifically to disrupted interactions with prefoldin, rather than general defects in VP3 function. To address this, we assessed early-stage viral replication in intestinal enteroids at 1 and 8 hours post-infection (hpi), time points when replication is less likely to be influenced by cumulative defects. As shown in the new Fig. 7b, we observed no significant differences in early replication between wild-type and VP3 mutant viruses, suggesting that the overall function of VP3 in the early phase of infection remains intact.

New Fig. 7b. VP3-671A/Y677A mutation virus is attenuated at late-stage infection in human enteroids.

Human small intestinal enteroids were infected with recombinant RVs expressing WT or RV-D671A/Y677A VP3 (MOI=0.1) for 1, 8, and 48 hours. Viral RNA was measured by RT-qPCR. Data represent the average of three experiments; error bars indicate SEM (two-way ANOVA with Tukey's post hoc test; ****, P<0.0001).

Reviewer #2:

Zhu et al describe the identification of a novel interaction between rotavirus VP3 and the prefoldin complex. This works aims to address the open questions about how VP3 can help to control immune response in rotavirus infected cells. These are very important and

timely questions; however the lack of experiment using virus infections, the lack of relevant cells models and several controls are critically missing to determine the authors claims. Please see below for specific comments:

A: We thank the reviewer for the positive summary of our work. In response to the concerns raised, we have conducted additional experiments to address key questions, including VP3 and PFDN3 interaction under infection conditions, *PFDN3* KO cells as additional validation in most of the experiments, IFN- β induced ISGs in *PFDN3* and *PFDN4* double KO cells, and IFN signaling blocking assays in *UBA3* KO cells. Please see the updated figures (new Fig. 1d, 2c, 3b, 3d, 6a, 6b, 6i, Supplementary Fig. 9, Supplementary Fig. 13, and Revision Fig. 2).

Major Points:

Q1. Overexpressing VP3 in HEK cells does not show real interactions in the context of infection. Need to show that VP3 interacts with PFDN3 when other rotavirus proteins are around. The authors mention that VP3 also interacts with MAVS and OAS. However, these proteins are missing from their pull-down (Fig. 1A). Does this mean that VP3 only interacts with these proteins during infection and not when expressed alone? The lack of known interactors being found with VP3 cause doubts that the HEK293 over expression system is mimicking interactions that occur during infection.

A1. We thank the reviewer for these insightful suggestions. To validate the VP3-PFDN3 interaction under infection conditions, we performed two new experiments. (1) We transfected GFP-VP3 into A549 cells and then super-infected them with rotavirus. Co-immunoprecipitation confirmed PFDN3 interaction with VP3 but not the GFP empty vector in an infected cell type that is interferon-competent (new Fig. 1d; lines 79-80).

New Fig. 1d. VP3 interacts with PFDN3 in rotavirus-infected A549 cells.

A549 cells were transfected with the indicated plasmids for 24 hours, followed by infection with rotavirus at an MOI of 1 for 12 hours. Co-immunoprecipitation was performed, and the interaction between VP3 and PFDN3 was analyzed by western blot.

(2) We rescued a recombinant rotavirus expressing a Flag-tagged VP3 GTase/RTPase domain (RV-GTase-Flag). The recombinant virus replicated comparably to the parental RV in MA104 cells, and the modified NSP3-P2A-GTase-Flag gene was confirmed by sequencing (new Supplementary Fig. 13a and b). Co-immunoprecipitation in infected HT-29 cells confirmed that the GTase domain interacted with endogenous PFDN3 (new Fig. 6i; lines 221–228).

New Fig. 6i. VP3-GTase domain interacts with PFDN3 during RV infection.

HT-29 cells were infected with wild-type or SA11 expressing VP3-GTase-Flag (MOI=0.1) for 24 hours. Cell lysates were subject to immunoprecipitation using α -Flag antibody and probed with α -PFDN3 antibody.

New Supplementary Fig. 13. Characterization of an RV expressing Flag-tagged VP3-GTase domain.

(a) MA104 cells were infected with either parental RV or RV-VP3-GTase-Flag virus at an MOI of 0.01. Virus titers were determined by an FFU assay at 1, 8, 24, and 48 hours post infection.

(b) The NSP3-P2A-VP3-GTase-Flag gene was amplified from virus stock and verified by Sanger sequencing.

Regarding the absence of MAVS and OAS in the HEK293 overexpression pulldown, we agree this likely reflects limitations of the system, as some interactions may only occur in the context of infection. We have clarified this point in the revised manuscript (lines 277-281).

Q2. In addition, HEK293T cells often lack many factors required to stimulate and induce ISGs. Would be important to show that interactions between VP3 and PFDN3/4/5 occur in more physiologically relevant cells with known intact immune sensing pathways.

A2. We thank the reviewer for this important suggestion. To address the concern regarding physiological relevance, we conducted additional experiments using human alveolar basal epithelial cell line (A549) and human colon cancer cell line (HT-29), both of which possess robust IFN signaling and other innate immune

pathways. In A549 cells, we transfected GFP-tagged VP3 and subsequently infected the cells with rotavirus. Co-immunoprecipitation analysis confirmed that VP3 interacts with PFDN3 under infection conditions (new Fig. 1d, lines 79-80). In addition, we performed co-immunoprecipitation assays in HT-29 cells, demonstrating that the Flag-tagged GTase domain of VP3 also interacts with PFDN3 during rotavirus infection (new Fig. 6i, lines 221-228).

New Fig. 1d. VP3 interacts with PFDN3 in rotavirus-infected A549 cells.

A549 cells were transfected with the indicated plasmids for 24 hours, followed by infection with rotavirus at an MOI of 1 for 12 hours. Co-immunoprecipitation was performed, and the interaction between VP3 and PFDN3 was analyzed by western blot.

New Fig. 6i VP3-GTase domain interacts with PFDN3 during RV infection.

HT-29 cells were infected with SA11-GTase-Flag virus for 24 hours. Cell lysates were subject to immunoprecipitation using α -Flag antibody and probed with α -PFDN3 antibody.

Q3. Confocal images in 1C and Sup 2 do not show any evidence of co-localization. The magnification is too low and they are both cytoplasmic which does not show that co-localization is occurring.

A3. We thank the reviewer for this valuable feedback, which was also raised by reviewer #1. We agree that the immunofluorescence data have limited value, as both proteins were overexpressed and diffusely cytoplasmic, which did not provide strong evidence for co-localization. Therefore, we have removed the immunofluorescence panel from the revised manuscript. Please also see response to Reviewer 1, Q1.

Q4. The authors show that *PFDN3* shows the greatest interaction with VP3. *PFDN3* KO cells were made however they were left out of many experiments. It is unclear why they were not used to test bovine RV in Fig. 2C. It was also not clear why they were excluded from the RNA-Seq analysis in Fig 3A, OAS3 analysis in Fig 3B, and MX1 analysis in Fig. 3D.

A4. We thank the reviewer for this comment. In response, we have now included *PFDN3* KO data in the viral RNA analysis (new Fig. 2c), OAS3 analysis (new Fig. 3b), and MX1 analysis (new Fig. 3d). Please see the new Fig. 2c, Fig. 3b and 3d.

New Fig. 2c. CRISPR knockout of *PFDN* promotes RV infection.

WT, *PFDN3* KO, and *PFDN4* KO HEK293 cells were infected with RV (UK strain, MOI = 5), and viral NSP5 mRNA level was measured at 24, 48, and 72 hpi by RT-qPCR. Data represent the average of three experiments; error bars indicate SEM (two-way ANOVA with Tukey's post hoc test; ***, $P < 0.001$; ****, $P < 0.0001$; ns, not significant).

New Fig. 3b. WT, *PFDN3* KO, and *PFDN4* KO cells were stimulated with IFN-β (500 U/ml), and ISGs (IFITM1 and MX1) mRNA level was measured at 24 hpi by RT-qPCR (left and middle panel). WT, *PFDN3* KO, *PFDN4* KO, and *PFDN4* rescue cells were stimulated with IFN-β (500 U/ml), and the OAS3 mRNA level was measured at 24 hpi by RT-qPCR (right panel). Data represent the average of three experiments; error bars indicate SEM (two-way ANOVA with Tukey's post hoc test; ****, $P < 0.0001$; ns, not significant).

New Fig. 3d. WT, *PFDN3* KO, and *PFDN4* KO HEK293 cells were infected with or without RV (RRV strain) at an MOI of 5 for 24 hours. MX1 mRNA level was measured at 24 hpi by RT-qPCR, which was normalized to GAPDH. Data represent the average of three experiments; error bars indicate SEM (two-way ANOVA with Tukey's post hoc test; ****, $P < 0.0001$).

Q5. The observation that PFDN3 cells also lose PFDN4 is very concerning. This makes any data shown with the PFDN3 KO cells hard to interpret. The authors need to show all PFDN components in their PFDN3 and PFDN4 knock-down cells to clarify this matter. Also, it would be important to show multiple clones of these KO cells. The authors have shown this for their UBA3 KO cells but did not do this for PFDN3 and PFDN4.

A5. We thank the reviewer for this valuable comment. The integrity of the prefoldin complex appears to depend on the presence of multiple subunits, and loss of a single component can destabilize the entire complex. Consistent with a previous report (PMID: 33137104), siRNA-mediated knockdown of *PFDN3* leads to decreased protein levels of other PFDN subunits, including *PFDN1*, *PFDN2*, *PFDN4*, and *PFDN5* (see Revision Fig. 2). Our *PFDN3* KO cells exhibit a similar pattern, suggesting that the observed downregulation of other subunits is a consequence of complex destabilization.

Revision Fig. 2 PFDN3 silencing revealed that other PFDN subunits were also down-regulated (PFDN1, PFDN2, PFDN4, and PFDN5) (PMID: 33137104).

In response to the reviewer’s request, we also validated the knockout efficiency and ISG expression profiles in multiple independent clones of both *PFDN3* and *PFDN4* KO cells. Western blot analysis confirmed robust depletion of the target proteins across clones, and ISG induction following IFN- β stimulation was comparable between clones (see new Supplementary Fig. 7).

Supplementary Fig. 7. ISG expression was reduced in different clones of *PFDN3* and *PFDN4* KO cells.

(a) WT and *PFDN3* (clones #1, #2) HEK293 cells lysates were subjected to western blot analysis. (b) WT and *PFDN4* (clones #1, #2) HEK293 cells lysates were subjected to western blot analysis. (c) WT, *PFDN3* KO (clones #1, #2), and *PFDN4* KO (clones #1, #2) HEK293 cells were stimulated with IFN- β (500 U/ml) for 24 hours. MX1 mRNA level was measured by RT-qPCR. Data represent the average of three experiments; error bars indicate SEM (two-way ANOVA with Tukey's post hoc test; ****, $P < 0.0001$).

Q5a. Can the authors re-overexpress *PFDN4* in their *PFDN3* KO cells to see if loss of *PFDN3* alone has a phenotype. In the current version of the manuscript, it is difficult to make any conclusions whether *PFDN3* is playing a role or whether all phenotypes are driven by *PFDN4* exclusively.

A5a. To further clarify the role of individual subunits, we generated double KO cells by knocking out *PFDN4* in the *PFDN3* KO background. IFN- β stimulation of these double KO cells did not result in further reduction of ISG expression compared to *PFDN3* KO alone, suggesting that the observed phenotypes are driven by the loss of the entire complex rather than an individual subunit (new Supplementary Fig. 9).

Supplementary Fig. 9. PFDN3 and PFDN4 double KO do not further inhibit ISG expression.

WT, PFDN3 KO, PFDN4 KO, and PFDN3 and PFDN4 double KO (DKO) cells were stimulated with IFN-β (500 U/ml), and MX1 mRNA level was measured at 24 hpi by RT-qPCR. Cell lysates were subjected to western blot analysis. Data represent the average of three experiments; error bars indicate SEM (two-way ANOVA with Tukey's post hoc test; ****, $P < 0.0001$, ns, not significant).

Q6. From Figure 1B, PFDN3 and PFDN5 were the proteins that were shown to interact the most with RV VP3 so it is unclear why PFDN5 was also not analyzed by KO and how PFDN4 was chosen.

A6. We thank the reviewer for this comment. The prefoldin complex is composed of multiple subunits, and knockout of any single subunit destabilizes the others. While PFDN5 interacts strongly with VP3, it is also involved in regulating expression of genes such as Myc (PMID: 11585818), which may complicate the interpretation. Thus, we focused on PFDN3 and PFDN4, with PFDN3 showing the strongest interaction with VP3.

Q7. Figure 4E is missing a control to show that another myc-tagged protein would also not interact with UBA3.

A7. We thank the reviewer for this helpful comment. In our experiment, we included the pCMV6-entry Myc-DDK empty vector as a negative control, which expresses the Myc tag but does not contain any target protein. This control was used to demonstrate that the interaction observed is specific to the protein of interest rather than due to the Myc tag itself. We have described this control in the revised manuscript (line 170).

Q8. In Figure 5C, the authors show that PFDN4 KO cells also have low levels of UBA3. How can the authors separate out these functions. From the current state, it seems like loss of UBA3 is the dominant phenotype and not loss of PFDN4.

A8. We thank the reviewer for this comment. The loss of PFDN4 leads to the degradation of UBA3 (Fig. 5c), suggesting that *PFDN4* KO likely results in UBA3 unfolding and degradation. The fact that *PFDN4* and *UBA3* double KO cells phenocopied either single KO cells (Fig. 5b) supports that the two proteins likely act in the same signaling pathway.

Q9. Figure 6 missing control to show the amount of RV infection and VP3 expression.

A9. We thank the reviewer for this comment. We have now added VP6 as a control to reflect active virus infection. Unfortunately, we do not have an antibody that readily detects VP3 protein in RV-infected cells. Please see the new Fig. 6a.

New Fig. 6a. VP3 competes with UBA3 for PFDN3 interaction.

PFDN4 KO HEK293 cells were transfected with PFDN1-6-Myc plasmids for 24 hours and infected with or without RV (RRV strain, MOI of 5) for another 24 hours. Cell lysates were subject to immunoprecipitation using α -Myc antibody and probed for α -UBA3 antibody. PFDN3 is the top band, and PFDN1-4 and PFDN6 are similar in size, as shown on the bottom.

Minor Points:

Q10. In 6B its not clear that these are truncated mutants and not point mutants. A better schematic is needed. Also, in this figure it seems like GFP alone is associating with PFDN3.

A10. We appreciate the reviewer's comment and apologize for the confusion. We have revised the schematic to clearly indicate the truncated mutants (new Fig. 6b). Additionally, GFP itself is not associated with PFDN3; the non-specific band is slightly larger than endogenous PFDN3 and is now marked with an asterisk (new Fig. 6c).

New Fig. 6b and 6c. GTase domain is essential for VP3 and PFDN3 interaction.

(b) Schematic diagram of WT and mutant VP3 proteins with defined domains illustrated in colors.

(c) HEK293 cells were transfected with GFP-tagged VP3 mutants for 48 hours. Cell lysates were subject to immunoprecipitation using α -GFP antibody and probed for α -PFDN3 antibody. * Represents a non-specific band.

Q11. No sure what MA104 SERPINB1 cells were used for – they were listed in Materials and Methods.

A11. We thank the reviewer for the comment, which was also asked by reviewer #3. *SERPINB1* KO MA104 cells were used in our rotavirus reverse genetics system and characterized in a recent publication from our lab (PMID: 39505878). We have added more detailed information about the reverse genetics system in the Materials and Methods section. Please see lines 314-315.

Reviewer #3:

The manuscript “Prefoldin complex promotes interferon-stimulated gene expression and is inhibited by rotavirus VP3” by Zhu et al. reports a novel prefoldin-UBA3-IRF9-ISG axis in antiviral innate immunity, and VP3 interfered this axis through direct and competitive binding to PFDNs. The authors identified PFDNs as a key protein family that interacts with VP3 by protein-protein interaction identification and reverse genetics, and revealed a novel role for PFDNs in antiviral immunity, which is of great significance in expanding not only the understanding of rotavirus VP3 antagonism to interferon signaling, but also new mechanisms in the field of antiviral resistance. However, there are still some revisions need to be clarified and the comments are as follows:

A: We thank the reviewer for the positive summary and thoughtful evaluation of our work. In response to the comments, we have conducted additional experiments to strengthen our findings and address the raised concerns. Specifically, we included *PFDN3* KO cells as additional validation in several key experiments, examined rotavirus replication at earlier time points, assessed viral replication in *PFDN4*-overexpressing cells, and evaluated ISG expression in *PFDN3/PFDN4* double KO cells. Furthermore, we performed *in vivo* experiments to support our conclusions. We have carefully revised the manuscript to incorporate these new findings, including new data in Fig. 2c, 2d, 3b, 3d, 7e, 7f, Supplementary Fig. 9, and Revision Fig. 3, 4.

Major:

*Q1. The authors mainly applied *PFDN3* and *PFDN4* from PFDNs to study, some experiments applied both *PFDN3* and *PFDN4*, some only applied one of them, what is the reasons for this? For example, according to the manuscript, VP3 directly interacts with *PFDN3* and knockout of *PFDN3* would simultaneously make *PFDN4* almost undetectable, so it seems that *PFDN3* is more significant, whereas the initial characterization of the role of PFDN in the RV (Fig.2C) and the effect on ISG (Fig.3A) just applied *PFDN4* but not *PFDN3*, what is the reason for this?*

A1. We thank the reviewer for this comment. In response, we have now included *PFDN3* KO data in the viral RNA analysis (new Fig. 2c), *OAS3* analysis (new Fig. 3b), and *MX1* analysis (new Fig. 3d). Please see the new Fig. 2c, Fig. 3b, and 3d.

New Fig. 2c. CRISPR knockout of *PFDN* promotes RV infection.

WT, *PFDN3* KO, and *PFDN4* KO HEK293 cells were infected with RV (UK strain, MOI = 5), and viral NSP5 mRNA level was measured at 24, 48, and 72 hpi by RT-qPCR. Data represent the average of three experiments; error bars indicate SEM (two-way ANOVA with Tukey's post hoc test; ***, P < 0.001; ****, P < 0.0001; ns, not significant).

New Fig. 3b. WT, *PFDN3* KO, and *PFDN4* KO cells were stimulated with IFN- β (500 U/ml), and ISGs (IFITM1 and MX1) mRNA level was measured at 24 hpi by RT-qPCR (left and middle panel). WT, *PFDN3* KO, *PFDN4* KO, and *PFDN4* rescue cells were stimulated with IFN- β (500 U/ml), and the *OAS3* mRNA level was measured at 24 hpi by RT-qPCR

(right panel). Data represents the average of three experiments; error bars indicate SEM (two-way ANOVA with Tukey's post hoc test; ****, $P < 0.0001$; ns, not significant).

d

New Fig. 3d. WT, *PFDN3* KO, and *PFDN4* KO HEK293 cells were infected with or without RV (RRV strain) at an MOI of 5 for 24 hours. MX1 mRNA level was measured at 24 hpi by RT-qPCR, which was normalized to GAPDH. Data represents the average of three experiments; error bars indicate SEM (two-way ANOVA with Tukey's post hoc test; ****, $P < 0.0001$).

Q2. The VP3 protein used for interactions in the text is from RRV, and when verifying the effect of PFDN on RV, UK, WI61, and OSU strains were applied, but not RRV, what is the reason for this?

A2. We thank the reviewer for the comment. We chose the bovine UK, human WI61, and porcine OSU strains for these experiments because they are IFN-sensitive, whereas RRV is less so, at least in human cell lines (PMID: 21307186 and 21994581). For reference, replication data comparing UK and RRV strains in WT and MAVS knockout cells is shown below in Revision Fig. 3.

Revision Fig. 3 UK is sensitive to IFN but not RRV (PMID: 21307186).

Q3. In the paper, there are only results of the effect on RV after PFDN knockout, is it possible to add some PFDN overexpression data?

A3. We thank the reviewer for the comment. In response, we have now added PFDN4 overexpression data. We found that overexpression PFDN4 only modestly inhibited RV replication (Revision Fig. 4). PFDN4-Myc expression was confirmed by western blot using anti-Myc and anti-PFDN4 antibodies. Please see Revision Fig. 4.

Revision Fig. 4 PFDN4 overexpression inhibited RV replication.

HEK293 cells were transfected with PFDN4-Myc plasmid and then infected with RV (MOI=5) for 24 hours. Virus titers were determined by an FFU assay. Data presents three dependent assay, error bars indicate SEM (student t-test; **, P<0.01). Cell lysates were subjected to western blot analysis.

Q4. As in Fig.2C, at 24 h, PFDN seems to be proviral (although not statistically significant), so would PFDN have a role in promoting rotavirus infection at earlier time points? Because PFDN affect actin polymerization, which may participate in RV entry.

A4. We thank the reviewer for this comment. We measured virus titers at earlier time points (1 hpi and 8 hpi) and found no significant difference between PFDN KO and WT cells, suggesting viral entry and other steps within one replication cycle were likely not altered by the presence or absence of PFDN. Please see the new Fig. 2d.

New Fig. 2d. CRISPR knockout of PFDN promotes RV replication at late time points

WT, PFDN3 KO, and PFDN4 KO HEK293 cells were infected with RV (UK strain, MOI = 5), and viral NSP5 mRNA level was measured at 1, 8, 24, 48, and 72 hpi by RT-qPCR. Data represent the average of three experiments; error bars indicate SEM (two-way ANOVA with Tukey's post hoc test; *, P<0.05; ****, P<0.0001; ns, not significant).

Q5. The text identifies ISGs expression mainly using MX1 as the indicator, and the main reason for choosing this indicator and related references should be added to the text.

A5. We thank the reviewer for the comment. MX1 is one of the most highly induced genes upon IFN stimulation (Fig. 3a). We have now added the reasoning for choosing MX1 as the indicator for ISG expression in the results section. Please see lines 130-131.

Q6. For PFDNs and UBA3, it is suggested that UBA3 interacted with mix of PFDNs plasmids (Fig.4E), and then demonstrated that UBA3 directly interacted with PFDN3. I wonder if any attempts were made to characterized the direct interactions of UBA3 with other PFDNs, which may affect the ability of VP3 to block the interactions of PFDNs and UBA3. Additionally, in the further experiments to characterize the role of UBA3 in ISG production, it was switched to PFDN4 KO cells (Fig.5), and the reason for this should be explained clearly in the text.

A6. Thank you for the thoughtful comment. We were unable to assess direct interactions between UBA3 and other PFDN subunits due to the lack of recombinant proteins for PFDN1, 2, 4, 5, and 6. Thus, we focused on PFDN3, for which reagents were readily available. For the ISG analysis in Fig. 5, we used PFDN4 knockout cells because they are the most extensively characterized in our system, including available RNA-seq data. We have clarified these points in the revised text. Please see lines 181-182.

Q7. What's relationship of PFDN3 and PFDN4, complementary, independent, or mutually enhancing? Did the author perform double KO experiment of PFDN3 and PFDN4.

A7. To determine the relationship between PFDN3 and PFDN4, we generated double KO cells by knocking out PFDN4 in PFDN3 KO cells. IFN- β stimulation assays using these double KO cells showed no further reduction in ISGs compared to PFDN3 KO alone, suggesting that PFDN3 and PFDN4 are not functionally complementary or mutually enhancing in this context. Notably, silencing of PFDN3 led to the downregulation of PFDN1, PFDN2, PFDN4, and PFDN5 (see Revision Fig. 2), indicating that PFDN3 KO alone already disrupts the stability of the entire prefoldin complex. Thus, single PFDN3 KO cells functionally resemble PFDN3/PFDN4 double KO cells. Please also refer to new Supplementary Fig. 9, lines 135-138, and response to Reviewer 2, Q5 for further detail.

Supplementary Fig. 9. PFDN3 and PFDN4 double KO do not further inhibit ISG expression.

(a) WT, PFDN3 KO, PFDN4 KO, and PFDN3 and PFDN4 double KO (DKO) cells were stimulated with IFN- β (500 U/ml) for 24 hours and cell lysates were subjected to western blot analysis. (b) MX1 mRNA level was measured at by RT-qPCR. Data represents the average of three experiments; error bars indicate SEM (two-way ANOVA with Tukey's post hoc test; ****, $P < 0.0001$, ns, not significant).

Q8. Additionally, what's exact function of PFDNs to UBA3? assists in folding (line 23), PFDN operates upstream of UBA3 (line 235), or PFDN stabilizes UBA3 (line 236)? These are different biological processes. And it's related to the Fig.8, the first figure of the working model.

A8. UBA3 RNA levels were not altered in PFDN KO cells (Fig. 3a). PFDN seems to modulate UBA3 protein levels by assisting its folding, based on our data and its traditionally reported function (PMID: 9463374; PMID: 12456645). But we do not have direct evidence that PFDN hands UBA3 off to the CCT complex.

Q9. In introduction, the authors mentioned that PFDN seems to be proviral in previous publications. But our results proved that PFDN possesses antiviral function? Why does this difference exist in different viruses? The authors should discuss this in the discussion part.

A9. We thank the reviewer for the suggestion. We have added more discussion about this point. Please see lines 296-299.

Q10. The *in vivo* experiments are important to the conclusion.

A10. We thank the reviewer for this important comment. We have now performed additional *in vivo* experiments as requested. Five-day-old C57BL/6 pups were orally infected with either WT or VP3-D671A/Y677A mutant virus for 24 hours. Small intestines were collected, and viral mRNA levels were measured. The results showed that the mutant virus exhibited attenuated replication *in vivo* compared to the parental virus with wild-type VP3. These data have been included in the revised manuscript (see new Fig. 7e–f, lines 252–253).

New Fig. 7e-f *In vivo* experiments demonstrated that the mutant VP3 exhibited reduced viral replication compared to the wild-type.

Five-day-old C57BL/6 pups were infected with either WT virus (n=9) or VP3-D671A/Y677A mutant virus (n=6) for 24 hours. Small intestinal tissues, i.e., duodenum (e) and jejunum (f) were collected, and viral mRNA levels were quantified by RT-qPCR. Data represents the mean of three independent experiments; error bars indicate SEM. Statistical significance was determined by Student's t-test (P < 0.05).

Other comments

Q11. Some words should be in its full name because it's appeared in the first time, such as RVs, MAVS, etc.. Please double check.

A11. Corrected as suggested (lines 46 and 51).

Q12. *For PFDN3, its current gene symbol is VBP1. Of course, there is not a problem with calling it PFDN3 to show that it is part of PFDNs, but it's better to explain in the text so as not to make confuse.*

A12. Corrected as suggested (line 33).

Q13. *Line 41: The reference here is maybe wrong, and it looks like the following:*

Mousnier A, Kubat N, Massias-Simon A, Ségéral E, Rain JC, Benarous R, Emiliani S, Dargemont C. von Hippel Lindau binding protein 1-mediated degradation of integrase affects HIV-1 gene expression at a postintegration step. Proc Natl Acad Sci U S A. 2007 Aug 21;104(34):13615-20. doi: 10.1073/pnas.0705162104. Epub 2007 Aug 13. PMID: 17698809

A13. We thank the reviewer for the comment. Corrected as suggested (line 42).

Q14. *Line 125: The figure here should be (Supplementary Fig.7B).*

A14. Corrected as suggested (line 130).

Q15. *Line 150-152: "...we treated cells with small-molecule inhibitors AGI6780 (33), MLN4924 (34), and Disulfiram (35), which block UBA3, IDH2, and ALDH2, respectively. Among the compounds tested, only MLN4924, an inhibitor of UBA3 and the neddylation pathway..." There is a contradiction of the sentence. In the first part, AGI6780 is inhibitors of UBA3; in the later part, it seems MLN4924 is inhibitor of UBA3. Please double check.*

A15. Corrected as suggested (lines 160-161).

Q16. *Line 154: Why do you think 100 nM is very low concentration? How about 1 or 10 nM?*

A16. We changed the word "very low" to "the lowest concentrations tested" (line 164).

Q17. *Line 173-174: UBA3 protein level was almost undetectable in PFDN4 knockout cells. Why did this happen? The author need to be explained somewhere.*

A17. UBA3 protein levels were reduced in both PFDN3 knockout and PFDN4 knockout cells compared to WT HEK293 cells (Fig. 4d), suggesting that UBA3 may

be a client protein of PFDNs. The loss of PFDN function might impair proper folding or stability of UBA3, leading to its decreased protein level. We have now added this explanation in the Results section (lines 165-167).

Q18. Line 233-234: *This sentence is a bit exaggerated, “We have demonstrated that knocking out individual PFDN subunits significantly enhances viral replication”. The authors only studied PFDN3 and 4, not each individual PFDN subunits.*

A18. Corrected as suggested (lines 260-261).

Q19. Line 256: *“... IRF9 through weather through neddylation modification...” is incorrect.*

A19. Corrected as suggested. Please see line 288.

Q20. Line 264-271: *The section is unnecessary.*

A20. We thank the reviewer for the comment. We agree, and we have removed it for simplicity.

Q21. Line 285: *I couldn't find the cell lines in the results, “MA104 SERPINB1 knockout cells”*

A21. We used this cell line to rescue recombinant RVs. We have added text and citation (PMID: 39505878). Please see lines 314-315 and the response to Reviewer 2, Q11.

Q22. Line 287: *The information of these RV strains need to be provided.*

A22. Corrected as suggested. Please see line 317.

Q23. Line 439: *The full stop should be deleted.*

A23. Corrected as suggested.

References

- 1 Chesnel, F. et al. The prefoldin complex stabilizes the von Hippel-Lindau protein against aggregation and degradation. *PLoS Genet* **16**, e1009183 (2020). <https://doi.org/10.1371/journal.pgen.1009183> (PMID: 33137104).
- 2 Satou, A., Taira, T., Iguchi-Arigo, S. M. M. & Ariga, H. A novel transrepression pathway of c-Myc. Recruitment of a transcriptional corepressor complex to c-Myc by MM-1, a c-Myc-binding protein. *Journal of Biological Chemistry* **276**, 46562-46567 (2001). <https://doi.org/10.1074/jbc.M104937200> (PMID: 11585818).
- 3 Zhu, Y. et al. CRISPR/Cas9 screens identify key host factors that enhance rotavirus reverse genetics efficacy and vaccine production. *NPJ Vaccines* **9**, 211 (2024). <https://doi.org/10.1038/s41541-024-01007-7> (PMID: 39505878).
- 4 Sen, A., Pruijssers, A. J., Dermody, T. S., García-Sastre, A. & Greenberg, H. B. The Early Interferon Response to Rotavirus Is Regulated by PKR and Depends on MAVS/IPS-1, RIG-I, MDA-5, and IRF3. *Journal of virology* **85**, 3717-3732 (2011). <https://doi.org/10.1128/Jvi.02634-10> (PMID: 21307186).
- 5 Arnold, M. M. & Patton, J. T. Rotavirus Antagonism of the Innate Immune Response. *Viruses-Basel* **1**, 1035-1056 (2009). <https://doi.org/10.3390/v1031035> (PMID: 21994581).
6. S. Geissler, K. Siegers, E. Schiebel, A novel protein complex promoting formation of functional alpha- and gamma-tubulin. *The EMBO journal* **17**, 952-966 (1998). (PMID: 9463374).
7. J. Martin-Benito et al., Structure of eukaryotic prefoldin and of its complexes with unfolded actin and the cytosolic chaperonin CCT. *The EMBO journal* **21**, 6377-6386 (2002). (PMID: 12456645)

Response to the reviewers' comments on *Nature Communications* manuscript (ID: NCOMMS-24-84308, Zhu Y et al., Prefoldin complex promotes interferon-stimulated gene expression and is inhibited by rotavirus VP3).

Below please find our point-by-point response to the reviewers' comments:

Reviewer #1:

Reviewer #1 (Remarks to the Author):

The authors have done a commendable job and have responded to the majority of comments in a satisfactory manner.

We thank the reviewer for the positive feedback.

However, one response remains unsatisfactory: the explanation regarding the replication phenotype of the mutant viruses. It is unclear whether this phenotype is due to VP3's ability to counteract the prefoldin complex or due to other functions of VP3. Although the authors have generated tools (e.g., knockout cells) to address this question directly, they have not utilized them. Furthermore, because viral replication during the early time points (1–8 hours) is minimal, I do not believe the data support a definitive conclusion about which step of infection is affected by the VP3 mutation.

We thank the reviewer for the constructive feedback. We acknowledge that our current data do not allow us to definitively determine whether the replication phenotype of the mutant viruses arises from VP3's antagonism of the prefoldin complex or from other functions of VP3. We have revised the manuscript to temper our conclusions accordingly and will pursue further studies using our existing knockout cell models and VP3 mutants with discrete functional defects to address this question in future work.

Reviewer #2 (Remarks to the Author):

The authors have addressed all comments from the first revision. My only concern is still lack of the infection model. They use an infection with over expression to show the interaction in this revised version. However, this is still a bit concerning as there is still so much protein being expressed that it may push interactions that are not normally present under real infection conditions.

We thank the reviewer for the insightful comment. In addition to the overexpression-based infection model, we also performed infections with a

recombinant virus expressing the GTase domain of VP3. We observed that the GTase domain interacts with PFDN during infection, supporting the physiological relevance of the interaction. Due to the lack of a suitable VP3 antibody for immunoprecipitation, we were unable to demonstrate the interaction between full-length VP3 and PFDN during infection. We have added these details to the revised manuscript.

Reviewer #3 (Remarks to the Author):

The authors have responded all my comments and concerns.

One more suggestion, it's better to integrate all the sequence of sgRNAs or primers into one table.

We thank the reviewer for the helpful suggestion. In the revised manuscript, we have consolidated all sgRNA and primer sequences into a single table for clarity and ease of reference.

Reviewer #4 (Remarks to the Author):

Thank you.